# StPR: Spatiotemporal Preservation and Routing for Exemplar-Free Video Class-Incremental Learning

**Huaijie Wang**[1], **De Cheng**[1], **Guozhang Li**[2], **Zhipeng Xu**[1]
**Lingfeng He**[1], **Jie Li**[1], **Nannan Wang**[1], **Xinbo Gao**[1*]
[1]Xidian University, Xi'an, China [2]Beijing Normal University, Beijing, China
`huaijie_wang@stu.xidian.edu.cn, dcheng@xidian.edu.cn`

## ABSTRACT

Video Class-Incremental Learning (VCIL) seeks to develop models that continuously learn new action categories over time without forgetting previously acquired knowledge. Unlike traditional Class-Incremental Learning (CIL), VCIL introduces the added complexity of spatiotemporal structures, making it particularly challenging to mitigate catastrophic forgetting while effectively capturing both frame-shared semantics and temporal dynamics. Existing approaches either rely on exemplar rehearsal, raising concerns over memory and privacy, or adapt static image-based methods that neglect temporal modeling. To address these limitations, we propose Spatiotemporal Preservation and Routing (StPR) mechanism, a unified and exemplar-free VCIL framework that explicitly disentangles and preserves spatiotemporal information. We begin by introducing Frame-Shared Semantics Distillation (FSSD), which identifies semantically stable and meaningful channels by jointly considering channel-wise sensitivity and classification contribution. By selectively regularizing these important semantic channels, FSSD preserves prior knowledge while allowing for adaptation. Building on this preserved semantic space, we further design a Temporal Decomposition-based Mixture-of-Experts (TD-MoE), which dynamically routes task-specific experts according to temporal dynamics, thereby enabling inference without task IDs or stored exemplars. Through the synergy of FSSD and TD-MoE, StPR progressively leverages spatial semantics and temporal dynamics, culminating in a unified, exemplar-free VCIL framework. Extensive experiments on UCF101, HMDB51, SSv2 and Kinetics400 show that our method outperforms existing baselines while offering improved interpretability and efficiency in VCIL.

## 1 INTRODUCTION

Class-Incremental Learning (CIL) Li & Hoiem (2017); Belouadah et al. (2021); De Lange et al. (2021); Masana et al. (2022); Zhang et al. (2024); Wang et al. (2025) is a critical direction in deep learning Spolaore & Wacziarg (2009); Jhuang et al. (2013); Cheng et al. (2024a); Xu et al. (2025); Cheng et al. (2026) that develops models capable of learning from a sequence of tasks without forgetting previous knowledge, recognizing an ever-growing set of classes without past task data or identifiers. A key challenge is catastrophic forgetting McCloskey & Cohen (1989); Ratcliff (1990), where new knowledge overwrites old. While well studied for images, extending CIL to videos: Video Class-Incremental Learning (VCIL) Park et al. (2021); Villa et al. (2022), remains underexplored. VCIL differs from CIL by requiring continual learning of new categories while modeling frame-shared semantics and temporal dependencies, unlike CIL's focus on static images. This spatiotemporal complexity is critical for understanding actions, motion, and scene dynamics in real-world applications like surveillance, driver monitoring, and robotics. Further, memory and privacy constraints often prohibit storing past data, demanding continual learning without rehearsal.

---

*Corresponding author.

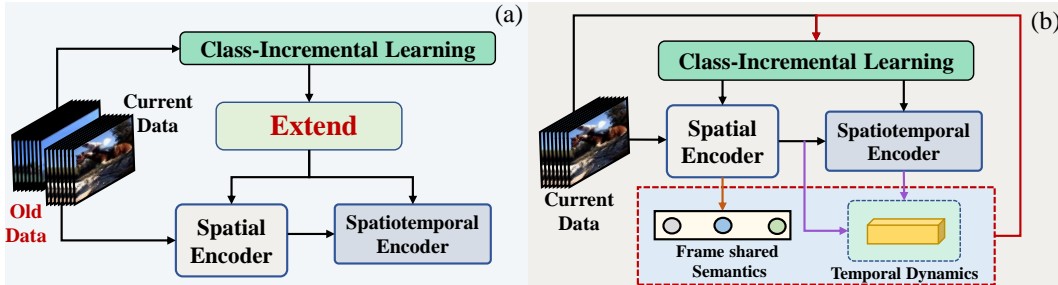

Figure 1: (a) Prior methods rely on exemplar rehearsal or naively stack video and CIL modules. (b) Our StPR framework explicitly decouples and reuses spatiotemporal semantics to mitigate forgetting.

The central challenge of VCIL lies in *mitigating catastrophic forgetting while effectively leveraging frame-shared semantics and temporal dynamics to incrementally learn new categories.* Existing methods can be broadly categorized into two types, as illustrated in Figure 1(a): 1) Exemplar-based methods Rebuffi et al. (2017); Hou et al. (2019); Douillard et al. (2020); Park et al. (2021); Pei et al. (2022); Villa et al. (2022); Alssum et al. (2023); Liang et al. (2024); Chen et al. (2025) store a portion of previous data (video clips, frames, or features) and apply rehearsal to reduce forgetting. However, storing exemplars incurs memory and privacy costs and typically emphasizes frame-level learning without explicitly modeling temporal dynamics. 2) CIL-based methods Li & Hoiem (2017); Dhar et al. (2019); Cheng et al. (2024b) adapt algorithms developed for static images, using techniques like regularization or subspace projection. While avoiding exemplar storage, they often overlook spatiotemporal properties by flattening or underutilizing temporal features. In contrast, our method StPR (Figure 1(b)) explicitly decouples video features into Frame shared semantics and temporal dynamics, and reuses these decomposed components to enhance the model's ability to adapt continually, thereby reducing forgetting without storing extensive exemplars.

Specifically, we propose a unified, exemplar-free VCIL framework that fully exploits the spatiotemporal nature of videos. Our method integrates both spatial semantic consistency and temporal variation to mitigate forgetting and improve adaptation across tasks. Separately, we introduce: 1) **Frame-Shared Semantics Distillation (FSSD).** To preserve frame-shared semantics and alleviate forgetting, we quantify the semantic importance of each channel using a combination of semantic sensitivity and classification contribution. This ensures that semantically meaningful and stable channels are preserved, achieving a better trade-off between stability and plasticity. 2) **Temporal-Decomposition-based Mixture-of-Experts (TD-MoE).** To exploit temporal dynamics for continual adaptation, we decouple task-specific temporal cues for each expert. At inference, expert routing depends solely on the temporal dynamics of the input, without requiring task identities or stored exemplars. This enables dynamic assignment of weights to experts according to the temporal dynamics of the input, facilitating incremental learning of new categories.

Our framework uniquely bridges the gap between video-specific spatiotemporal representation and class-incremental adaptation. By disentangling and leveraging both spatial semantic channel consistency and temporal dynamics, it offers an effective and explainable solution for continual video understanding. Our main contributions are: 1) We propose a Frame-Shared Semantics Distillation method (FSSD) that preserves frame-shared, semantically aligned spatial channels through semantic importance-aware regularization, optimizing the stability-plasticity trade-off in continual learningl; 2) We design a Temporal Decomposition based Mixture-of-Experts strategy (TD-MoE) that decomposes spatiotemporal features and uses temporal dynamics for expert combination, enabling task-id-free and dynamic adaptation; 3) We present a unified, exemplar-free VCIL framework that achieves state-of-the-art results on UCF101, HMDB51, SSv2 and Kinetics400, demonstrating the effectiveness of integrating spatial semantics and temporal dynamics in VCIL.

## 2 RELATED WORK

### 2.1 CLASS-INCREMENTAL LEARNING

Class-Incremental Learning (CIL) aims to enable models to continually learn new classes without forgetting previously learned ones. Existing approaches typically fall into three categories: (1)

*regularization-based methods* Kirkpatrick et al. (2017); Zenke et al. (2017); Xiang et al. (2022); Zhou et al. (2023); Li et al. (2024a); Wang et al. (2025), which constrain parameter updates to preserve prior knowledge, often via knowledge distillation Li & Hoiem (2017); Hou et al. (2019); (2) *exemplar-based methods* Bang et al. (2021); Chaudhry et al. (2018); Rebuffi et al. (2017), which store or generate past data to reduce forgetting; and (3) *structure-based methods* Serra et al. (2018); Mallya & Lazebnik (2018); Mallya et al. (2018); Liang & Li (2024); Yu et al. (2024); He et al. (2025b), which expand model capacity or isolate task-specific components. Recently, CIL combined with pre-trained vision transformers (ViTs) Ermis et al. (2022); Smith et al. (2023); Wang et al. (2022b;c); He et al. (2025a) has achieved impressive results by leveraging transferable representations and modularity. Some methods fully fine-tune pre-trained backbones Boschini et al. (2022); Zhang et al. (2023), but this is computationally expensive. To address efficiency, parameter-efficient fine-tuning (PEFT) methods have been introduced. Prompt pool-based approaches Wang et al. (2022c); Smith et al. (2023); Wang et al. (2024); Zhang et al. (2023) maintain task-specific prompts, while adapter-based methods Zhou et al. (2024a); Tan et al. (2024); Gao et al. (2024); Liang & Li (2024); Zhou et al. (2024b) adapt ViTs to new classes with minimal updates. While effective, most CIL strategies were originally developed for static image domains and do not generalize well to video-based scenarios, where temporal dynamics play a critical role.

## 2.2 Video Class-Incremental Learning

Action recognition has been widely explored with 2D CNNs using temporal aggregation Lin et al. (2019); Wang et al. (2016) and 3D CNNs or transformer for joint spatiotemporal modeling Carreira & Zisserman (2017); Tran et al. (2015); Li et al. (2024b). More recent work focuses on improving temporal sensitivity and efficiency Feichtenhofer (2020); Fan et al. (2020); Li et al. (2021) or text information Li et al. (2023). However, these models are trained in static setups and do not address continual adaptation or forgetting. To address these challenges, Video Class-Incremental Learning (VCIL) extends conventional Class-Incremental Learning (CIL) to spatiotemporal data, introducing additional challenges such as managing temporal variations across tasks. Several recent methods, including TCD Park et al. (2021), FrameMaker Pei et al. (2022), and HCE Liang et al. (2024), address this setting by storing videos or compressed exemplars. However, these strategies raise concerns related to memory efficiency and data privacy. While SMILE Alssum et al. (2023) effectively extracts image features from individual frames, it does not explicitly capture temporal information, which may limit its ability to leverage the distinctive decision cues present in video data. Exemplar-free methods such as STSP Cheng et al. (2024b) mitigate forgetting via orthogonal subspace projections, but they mainly adapt image-domain strategies to video tasks. In contrast, our approach decouples and models the spatiotemporal structure of videos, proposing a unified VCIL framework that preserves spatial consistency via Frame-Shared Semantics Distillation (FSSD) for knowledge retention without exemplars, while leveraging temporal dynamics for expert routing through Temporal Decomposition-based Mixture-of-Experts (TD-MoE).

## 3 Method

**Problem Definition**: In the Video Class-Incremental Learning (VCIL) setting, a model is trained across $B$ stages with sequentially arriving datasets $\{\mathcal{D}^1, \ldots, \mathcal{D}^B\}$. Each dataset $\mathcal{D}^b = \{(V_j^b, y_j^b)\}_{j=1}^{|\mathcal{D}^b|}$ corresponds to the $b$-th task, where $V_j^b$ is the $j$-th video and $y_j^b$ is its class label. Here, videos primarily represent human action recognition scenarios, where the spatiotemporal dynamics capture motion patterns, subject interactions, and scene context. $|\mathcal{D}^b|$ represents the number of samples in the $b$-th task. Let $\mathcal{Y}^b$ be the label space of the $b$-th dataset. For all $b \neq b'$, the label spaces are disjoint: $\mathcal{Y}^b \cap \mathcal{Y}^{b'} = \varnothing$. The objective of VCIL is to incrementally train a model over $B$ tasks while maintaining high performance across all accumulated classes $\{\mathcal{Y}^1, \mathcal{Y}^2, \ldots, \mathcal{Y}^B\}$.

**Overall framework.** We propose a unified, exemplar-free framework for Video Class-Incremental Learning (VCIL) built upon the CLIP model Radford et al. (2021). Our goal is to mitigate catastrophic forgetting while effectively leveraging frame-shared semantics and temporal dynamics to incrementally learn new categories. The frozen visual encoder $\mathcal{F}(\cdot)$ extracts spatial features, while adapters $\mathcal{A}^b$ are updated for each task $b$. A spatiotemporal encoder $\mathcal{G}(\cdot)$ models temporal dynamics. Our framework introduces two key components: 1) Frame-Shared Semantics Distillation (FSSD) identifies semantically stable channels across frames by combining Semantic Sensitivity and Classifi-

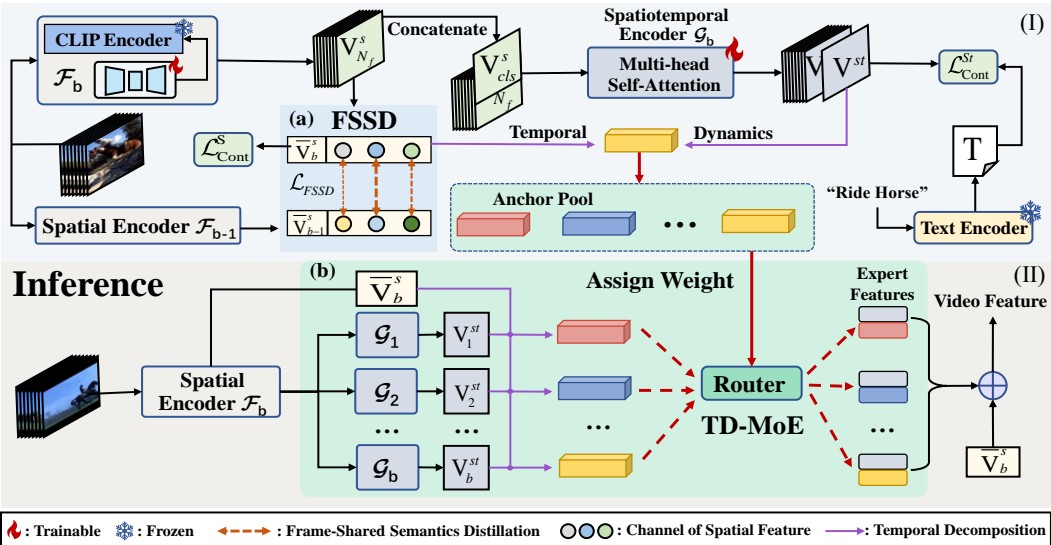

Figure 2: Overview of our proposed framework. (I) During training, the **(a) Frame-Shared Semantics Distillation (FSSD)** module retains past knowledge via frame-shared semantics importance-aware regularization. (II) At inference, the **(b) Temporal Decomposition-based Mixture-of-Experts (TD-MoE)** dynamically routes input videos to expert branches by leveraging the disentangled temporal component of the spatiotemporal representation, enabling adaptive predictions.

cation Score, applying selective regularization to preserve critical spatial semantics while maintaining plasticity. 2) Temporal Decomposition based Mixture-of-Experts (TD-MoE). To exploit temporal dynamics for continual adaptation, we decouple shared static components and temporal dynamics. During inference, temporal dynamics are used to assign dynamic weights to expert temporal encoders, enabling task-id-free adaptation without requiring task identifiers or stored exemplars.

## 3.1 SPATIAL AND SPATIOTEMPORAL ENCODER

**Spatial Encoder.** The shared adapter module Chen et al. (2022) $\mathcal{A}^b = \{\mathcal{A}_l^b\}_{l=1}^N$ is utilized with a frozen CLIP-ViT model with $N$ layers of transformer module, serving as the spatial extractor. An adapter is an encoder-decoder architecture embedded into the residual of each transformer layer, facilitating transfer learning and enhancing downstream task performance. Typically, it consists of a down-sampling MLP $\mathbf{W}_{down} \in \mathbb{R}^{d \times d_h}$, a ReLU activation $\phi(\cdot)$, and an up-sampling MLP $\mathbf{W}_{up} \in \mathbb{R}^{d_h \times d}$, where $d$ is the input and output dimension, and $d_h$ is the hidden dimension. For adapter, if input is $\mathbf{v}_i' \in \mathbb{R}^d$, which is the $i$-th frame-level feature after the Multi-Head Self-Attention and residual connection in CLIP ViT, the output of adapter is:

$$\mathbf{v}_i^a = \phi(\mathbf{v}_i' \mathbf{W}_{down}) \mathbf{W}_{up}. \tag{1}$$

The spatial feature of $i$-th frame $\mathbf{V}_i^s = \mathcal{F}(V_i; \mathcal{A}^b), \mathbf{V}_i^s \in \mathbb{R}^{d_{vt}}$, where $V_i$ is the $i$-th frame of the video, $d_{vt}$ is the dimension of the aligned video-text features.

**Spatiotemporal Encoder.** To obtain the spatiotemporal representation, we feed both the frame-level spatial features and a learnable [CLS] token into a multi-head self-attention based spatiotemporal encoder $\mathcal{G}(\cdot)$. Specifically, the input to $\mathcal{G}$ consists of the frame features $\mathbf{V}_1^s, \ldots, \mathbf{V}_{N_f}^s$ and a [CLS] token $\mathbf{V}_{cls}^s \in \mathbb{R}^{d_{vt}}$, where $N_f$ represents the number of sampled frames. The temporal encoder outputs the spatiotemporal feature $\mathbf{V}^{st}$, corresponding to the transformed [CLS] token:

$$\mathbf{V}^{st} = \mathcal{G}([\mathbf{V}_{cls}^s; \mathbf{V}_1^s; \ldots; \mathbf{V}_{N_f}^s])[0] \in \mathbb{R}^{d_{vt}}, \tag{2}$$

where [0] selects the output associated with the [CLS] token after attention-based aggregation.

## 3.2 FRAME-SHARED SEMANTICS DISTILLATION

In VCIL, shared adapter modules inevitably drift in feature space when adapting to new tasks, leading to forgetting. Directly applying classic uniform-weighted distillation from CIL to video tasks ignores the differences in semantic importance and temporal stability across video features. This leads to a suboptimal balance between stability and plasticity. To address this, we propose Frame-Shared Semantics Distillation (FSSD), which identifies stable cross-frame channels capturing core semantics and regularizes them to preserve prior knowledge while allowing adaptation

**Frame-Shared Semantics Distillation (FSSD).** To mitigate semantic drift across tasks, we introduce a distillation loss weighted by the frame-shared semantics importance:

$$\mathcal{L}_{FSSD} = \frac{1}{|\mathcal{D}^b| \cdot d_{vt}} \sum_c^{|\mathcal{C}_b|} \sum_i^{N_c} \sum_j^{d_{vt}} I_{b-1,c,j} \cdot \|\bar{V}_{b-1,c,i,j}^s - \bar{V}_{b,c,i,j}^s\|_2^2, \tag{3}$$

where $|\mathcal{C}_b|$ is the number of classes in the $b$-th task, and $N_c$ is the number of samples per class. $I_{b-1,c,j}$ denotes the frame-shared importance of the $j$-th channel for class $c$ from task $(b-1)$. $\bar{V}_{b-1,c,i,j}^s$ and $\bar{V}_{b,c,i,j}^s$ are the $j$-th channel outputs of the $i$-th sample in class $c$, extracted from the spatial encoders of task $(b-1)$ and $b$, respectively, calculated on current data, with the previous model frozen.

**Frame-Shared Semantics.** To quantify the importance of frame-shared semantics, we assess each channel based on two criteria: 1). **Semantic Sensitivity**. It measures the responsiveness to activation changes, thereby reflecting its reliability in representing consistent semantic information. and 2) **Classification Score**. It reflects the channel's contribution to the final classification.

For semantic sensitivity, we employ Fisher Information to estimate how sensitively a channel's activation influences the output. As spatial features $\bar{\mathbf{V}}^s = \frac{1}{N_f} \sum_{i=1}^{N_f} \mathbf{V}_i^s$ aggregate frame-wise variations, the Central Limit Theorem suggests each channel's distribution approximates a Gaussian (Belong to the same category). Thus, we assume the $j$-th channel activation for class $c$ follows (For simplicity, we omit the subscripts for task and sample.):

$$\bar{V}_{c,j}^s \sim \mathcal{N}(\mu_{c,j}, \sigma_{c,j}^2), \tag{4}$$

where $\mu_{c,j}$ and $\sigma_{c,j}^2$ denote the mean and variance across frames. The Fisher Information $\mathcal{I}(\mu_{c,j})$ (Detailed derivations are provided in the appendix B.1) with respect to $\mu_{c,j}$ is computed as:

$$\mathcal{I}(\mu_{c,j}) = \mathbb{E}\left[\left(\frac{\bar{V}_{c,j}^s - \mu_{c,j}}{\sigma_{c,j}^2}\right)^2\right] = \frac{1}{\sigma_{c,j}^4} \mathbb{E}\left[(\bar{V}_{c,j}^s - \mu_{c,j})^2\right] = \frac{1}{\sigma_{c,j}^4} \cdot \sigma_{c,j}^2 = \frac{1}{\sigma_{c,j}^2}. \tag{5}$$

For classification score, we compute the cosine similarity between the spatial video feature $\bar{\mathbf{V}}_c^s \in \mathbb{R}^{d_{vt}}$ and its corresponding text feature $\mathbf{T}_c \in \mathbb{R}^{d_{vt}}$. Specifically, for the $j$-th channel, the classification score is defined as $\gamma_{c,j} = \frac{\bar{V}_{c,j}^s \cdot T_{c,j}}{\|\bar{\mathbf{V}}_c^s\| \cdot \|\mathbf{T}_c\|}$, where $T_{c,j}$ denotes the $j$-th dimension feature of $\mathbf{T}_c$. We then take the expectation of $\gamma_{c,j}$ across frames to obtain a stable channel-level contribution estimate:

$$\mathbb{E}[\gamma_{c,j}] = \mathbb{E}\left[\frac{\bar{V}_{c,j}^s \cdot T_{c,j}}{\|\bar{\mathbf{V}}_c^s\| \cdot \|\mathbf{T}_c\|}\right] \propto \mathbb{E}\left[\frac{\bar{V}_{c,j}^s \cdot T_{c,j}}{\|\bar{\mathbf{V}}_c^s\|}\right] \approx \frac{T_{c,j} \cdot \mu_{c,j}}{\lambda}, \tag{6}$$

where $\|\bar{\mathbf{V}}_c^s\| \approx \lambda$ is treated as a constant after normalization.

Combining semantic sensitivity and classification score, the semantic importance for the $j$-th channel of the $c$-th class is defined as:

$$I_{c,j} = \frac{T_{c,j} \cdot \mu_{c,j}}{\sigma_{c,j}^2}. \tag{7}$$

FSSD accumulates frame-shared semantic importance as distillation weights, retaining key channels for old tasks while allowing less important ones to adapt, thus balancing stability and plasticity.

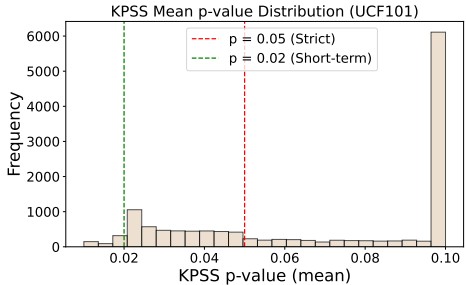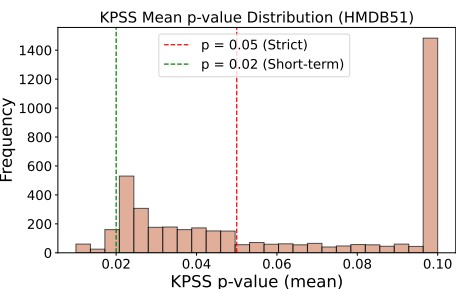

Figure 3: For each video, we uniformly sample 8 frames to compute the $p$-value defining $p > 0.05$ as strictly stationary and $p > 0.02$ as weakly stationary in the short term.

### 3.3 TEMPORAL DECOMPOSITION BASED MIXTURE-OF-EXPERTS

Given the high forgetting tendency of deep transformers in VCIL, we allocate a dedicated spatiotemporal encoder for each task. As task IDs are unavailable during inference, we allocate a spatiotemporal encoder per task and design a routing mechanism that dynamically weights experts based on temporal patterns, ensuring relevant experts contribute more to the final representation.

**Task-Specific Expert.** For each task, we train a dedicated expert based on the spatiotemporal encoder. The spatiotemporal features $\mathbf{V}^{st} \in \mathbb{R}^{d_{vt}}$ captured by each expert are computed as in Eq. 2.

**Temporal Decomposition-based Router.**

To design this routing mechanism based on temporal dynamics, we consider two aspects: 1) **Temporal residuals**. These reflect the subtle temporal differences within redundant frames. 2) **Inter-frame information.** This captures abstract temporal concepts between frames, based on the knowledge learned by each expert.

For temporal residuals, we observe that redundant frames, where backgrounds and subjects remain consistent, cause minimal variation between adjacent frames Kim & Choi (2024); Liu et al. (2021). This leads to short-term temporal stationarity, which we further validate on the UCF101 and HMDB51 datasets Fig. 3. Thus, each frame feature is decomposed as $\mathbf{V}_i^s = \bar{\mathbf{v}} + \boldsymbol{\epsilon}_i$, with $\bar{\mathbf{v}}$ as shared static components and $\boldsymbol{\epsilon}_i$ as temporal residuals. The spatial representation is then the mean across frames:

$$\bar{\mathbf{V}}^s = \bar{\mathbf{v}} + \bar{\boldsymbol{\epsilon}}, \quad \bar{\boldsymbol{\epsilon}} = \frac{1}{N_f} \sum_{i=1}^{N_f} \boldsymbol{\epsilon}_i. \tag{8}$$

For the inter-frame information, since the spatiotemporal feature $\mathbf{V}^{st}$ is computed by the attention module, it can be approximated as $\mathbf{V}^{st} \approx \sum_{i=1}^{N_f} a_i \cdot \mathbf{V}_i^s$, where $a_i$ is attention score. After normalization, we can obtain $\sum_{i=1}^{N_f} a_i = 1$. Substituting Eq. 8, we obtain:

$$\mathbf{V}^{st} = \bar{\mathbf{v}} + \sum_{i=1}^{N_f} a_i \cdot \boldsymbol{\epsilon}_i. \tag{9}$$

Since $\bar{\mathbf{v}}$ is difficult to estimate, and to decouple the temporal residual $\boldsymbol{\epsilon}_i$ and inter-frame information $a_i$, we naturally address this by using the difference between $\mathbf{V}^{st}$ and $\bar{\mathbf{V}}^s$, effectively isolating the temporal dynamics, which can be represented as:

$$\mathbf{V}^{tem} = \sum_{i=1}^{N_f} \left( a_i - \frac{1}{N_f} \right) \cdot \boldsymbol{\epsilon}_i. \tag{10}$$

This formulation reveals that $\mathbf{V}^{tem}$ quantifies the deviation between the model's attention-weighted temporal dynamics and the uniform temporal mean, effectively disentangling temporal variations from static semantics. This enables routing to exploit temporal cues while avoiding background interference, thereby mitigating forgetting and enhancing continual learning.

**Inference.** During inference, we first compute the decoupled temporal representation $\mathbf{V}^{tem} \in \mathbb{R}^{d_{vt}}$ for each input video. For all categories in the current task, we calculate the mean temporal representation and store it in the anchor pool as $\bar{\mathbf{V}}_c^{tem} \in \mathbb{R}^{d_{vt}}$, where $c$ represents the $c$-th class. For each expert $k$, we compute a similarity-based score as the router:

$$r_k = \max_{c \in \mathcal{C}_k} \cos\left(\mathbf{V}_k^{tem}, \bar{\mathbf{V}}_c^{tem}\right), \tag{11}$$

where $\mathcal{C}_k$ represents the set of classes assigned to expert $k$. Then, we combine the adapter-tuned spatial features $\bar{\mathbf{V}}^s$ with the expert outputs weighted by $r_k$ as the final video representation:

$$\mathbf{V} = \bar{\mathbf{V}}^s + \sum_k r_k \cdot \mathbf{V}_k^{st}. \tag{12}$$

The final video representation is matched with text embedding via cosine similarity for classification.

### 3.4 LOSS FUNCTION AND OPTIMIZATION

Our loss function includes: 1) contrastive loss between video features and text descriptions for classification; 2) contrast loss between video features under adapter fine-tuning and text features for spatial optimization; and 3) FSSD loss to mitigate forgetting in shared adapter modules.

**Contrastive Loss Formulation.** We use symmetric contrastive loss for video-to-text and text-to-video alignment. Given a batch of $N$ samples, let $\mathbf{V}_i$ and $\mathbf{T}_j$ denote the video and text features, respectively. The similarity between video $i$ and text $j$ is computed as the cosine similarity $S_{i,j} = \cos(\mathbf{V}_i, \mathbf{T}_j)$, forming a similarity matrix $S \in \mathbb{R}^{N \times N}$. Let $\mathbf{M} \in \{0,1\}^{N \times N}$ be the label mask, where $M_{i,j} = 1$ if $y_i = y_j$ and $M_{i,j} = 0$ otherwise. Then, the Video-to-text contrastive loss can be caculated by:

$$\mathcal{L}_{\text{v2t}} = -\frac{1}{N} \sum_{i=1}^{N} \log\left(\frac{\sum_{j=1}^{N} M_{i,j} \cdot \exp(S_{i,j})}{\sum_{j=1}^{N} \exp(S_{i,j}) + \varepsilon}\right), \tag{13}$$

where $\varepsilon$ is a small constant added to avoid division by zero. The Text-to-Video loss $\mathcal{L}_{\text{t2v}}$ is similarly defined by swapping video and text in the equation. Symmetric total contrastive loss is :

$$\mathcal{L}_{\text{Cont}} = \frac{1}{2}(\mathcal{L}_{\text{v2t}} + \mathcal{L}_{\text{t2v}}). \tag{14}$$

For $\mathcal{L}_{\text{Cont}}^{\text{St}}$, the embeddings are the spatiotemporal video feature $\mathbf{V}^{st}$ and corresponding text features. For $\mathcal{L}_{\text{Cont}}^{\text{S}}$, we use the CLIP adapter feature $\bar{\mathbf{V}}^s$ and corresponding text features.

**Total Loss.** The overall training loss is:

$$\mathcal{L} = \mathcal{L}_{\text{Cont}}^{\text{St}} + \mathcal{L}_{\text{Cont}}^{\text{S}} + w \cdot \mathcal{L}_{\text{FSSD}}, \tag{15}$$

where $w$ is a hyperparameter. This design aligns both spatial and spatiotemporal semantics with text supervision, while the FSSD term preserves critical frame-shared semantics to mitigate forgetting.

## 4 EXPERIMENTS

### 4.1 EXPERIMENTAL DETAILS

**Dataset.** We evaluate our method on four benchmark datasets: UCF101 Soomro et al. (2012), HMDB51 Kuehne et al. (2011), Something-Something V2 (SSv2) Goyal et al. (2017) and Kinetics400 Carreira & Zisserman (2017). All experiments are conducted in an exemplar-free setting. For fair comparison, we use the TCD benchmark Park et al. (2021) on UCF101, HMDB51, and SSv2, pretraining the model on 51, 26, and 84 base classes, respectively, with the remaining classes split into tasks. For Kinetics-400, we follow the vCLIMB benchmark Villa et al. (2022) with 10- or 20-task splits, each containing the same number of classes.

**Evaluation Metrics.** We adopt three widely-used metrics to evaluate performance in VCIL: 1). Final Accuracy (Acc) Villa et al. (2022), which measures the overall classification accuracy on all learned classes after the final task is completed; 2). Average Accuracy ($\overline{\text{Acc}}$) Park et al. (2021), which

Table 1: Average Accuracy ($\overline{\text{Acc}}$) of the UCF101 and HMDB51 under the TCD benchmark.

| Method | Exemplar | Venue | UCF101 | | | HMDB51 | |
| --- | --- | --- | --- | --- | --- | --- | --- |
| | | | $10 \times 5$s | $5 \times 10$s | $2 \times 25$s | $5 \times 5$s | $1 \times 25$s |
| iCaRL Rebuffi et al. (2017) | ✓ | CVPR'17 | 65.34 | 64.51 | 58.73 | 40.09 | 33.77 |
| LwFMC Li & Hoiem (2017) | ✗ | TPAMI'18 | 42.14 | 25.59 | 11.68 | 26.82 | 16.49 |
| LwM Dhar et al. (2019) | ✗ | CVPR'19 | 43.39 | 26.07 | 12.08 | 26.97 | 16.50 |
| UCIR Hou et al. (2019) | ✓ | CVPR'19 | 74.09 | 70.50 | 64.00 | 46.53 | 37.15 |
| PODNet Douillard et al. (2020) | ✓ | ECCV'20 | 74.37 | 73.75 | 71.87 | 48.78 | 46.62 |
| TCD Park et al. (2021) | ✓ | ICCV'21 | 77.16 | 75.35 | 74.01 | 50.36 | 46.66 |
| FrameMaker Pei et al. (2022) | ✓ | NeurIPS'22 | 78.64 | 78.14 | 77.49 | 51.12 | 47.37 |
| L2P Wang et al. (2022c) | ✗ | CVPR'22 | 81.24 | 80.09 | 78.58 | 49.98 | 45.87 |
| S-iPrompts Wang et al. (2022a) | ✗ | NeurIPS'22 | 80.60 | 80.27 | 80.43 | 53.11 | 53.89 |
| ST-Prompt†Pei et al. (2023) | ✗ | CVPR'23 | 84.75 | 85.54 | 85.67 | 60.14 | 60.54 |
| STSP Cheng et al. (2024b) | ✗ | ECCV'24 | 81.15 | 82.84 | 79.25 | 56.99 | 49.19 |
| HCE Liang et al. (2024) | ✓ | AAAI'24 | 80.01 | 78.81 | 77.62 | 52.01 | 48.94 |
| CoSTEO Zou et al. (2025) | ✓ | CVPR'25 | 86.05 | 86.71 | 86.95 | 61.70 | 61.84 |
| **StPR (Ours)** | ✗ | − | **94.67** | **92.13** | **88.52** | **68.12** | **67.01** |

Table 2: Average Accuracy ($\overline{\text{Acc}}$) of the SSv2 under the TCD benchmark, with best results in bold.

| Method | Exemplar | Venue | $10 \times 9$s | $5 \times 18$s |
| --- | --- | --- | --- | --- |
| iCaRL Rebuffi et al. (2017) | ✓ | CVPR'17 | 20.41 | 16.62 |
| UCIR Hou et al. (2019) | ✓ | CVPR'19 | 24.32 | 19.31 |
| PODNet Douillard et al. (2020) | ✓ | ECCV'20 | 27.63 | 20.14 |
| TCD Park et al. (2021) | ✓ | ICCV'21 | 29.32 | 24.69 |
| FrameMaker Pei et al. (2022) | ✓ | NeurIPS'22 | 31.41 | 26.57 |
| L2P Wang et al. (2022c) | ✗ | CVPR'22 | 26.02 | 21.33 |
| S-iPrompts Wang et al. (2022a) | ✗ | NeurIPS'22 | 33.69 | 30.84 |
| ST-Prompt†Pei et al. (2023) | ✗ | CVPR'23 | 39.98 | 35.44 |
| HCE Liang et al. (2024) | ✓ | AAAI'24 | 36.88 | 32.82 |
| CoSTEO Zou et al. (2025) | ✓ | CVPR'25 | **41.44** | 36.60 |
| **StPR (Ours)** | ✗ | − | 40.79 | **37.30** |

measures the mean classification accuracy over all incremental stages after the final task is completed; 3). Backward Forgetting (BWF) Villa et al. (2022), which quantifies the average drop in performance on previously learned tasks, reflecting how well the model retains past knowledge.

**Implementation Details.** All experiments are conducted on a single NVIDIA RTX 3090 GPU. We adopt the CLIP ViT-B/16 model Radford et al. (2021) as the backbone, with all its parameters frozen during training. The spatial and spatiotemporal encoders are the only trainable components in our framework. For optimization, we employ Stochastic Gradient Descent (SGD) with an initial learning rate of 0.01 and a batch size of 40. Each task is trained for 60 epochs in the first incremental session and 30 epochs in each subsequent session. The weighting hyperparameter $w$ in Eq. 15 is set to $1 \times 10^4$. The multi-head self-attention module within the spatiotemporal encoder consists of three transformer layers, each employing two attention heads. Video clips are sampled using the TSN strategy Wang et al. (2018), selecting 8 frames per video uniformly across the temporal dimension.

## 4.2 Main Results

Table. 1, 2 and 3 report results on UCF101, HMDB51, SSv2 and Kinetics400, covering different action complexities and temporal dynamics. Based on their strategies to mitigate forgetting, existing methods are categorized into two groups: 1) Exemplar-based methods (iCaRL, UCIR, PODNet, TCD, FrameMaker, HCE, vCLIMB, SMILE, CSTA). They store video clips, frames, or compressed features and apply rehearsal to reduce forgetting. However, these methods face scalability and privacy challenges due to their reliance on stored exemplars. 2) CIL-based methods (LwFMC, LwM, L2P, S-iPrompts, ST-Prompt†, STSP). This group adapts techniques from image-based class-incremental learning, such as unified distillation and subspace projection, without storing exemplars. While avoiding exemplar storage, their performance tends to be lower, especially as task difficulty increases and lacking explainable spatiotemporal disentanglement. In contrast, Our method (StPR) without

Table 3: Results of the Kinetics-400 under the vCLIMB benchmark at 10 and 20 task settings.

| Method | Exemplars | Venue | Kinetics400-10s Acc ↑ | Kinetics400-10s BWF ↓ | Kinetics400-20s Acc ↑ | Kinetics400-20s BWF ↓ |
|---|---|---|---|---|---|---|
| vCLIMB+BiC Villa et al. (2022) | ✓ | CVPR'22 | 27.90 | 51.96 | 23.06 | 58.97 |
| vCLIMB+iCaRL Villa et al. (2022) | ✓ | CVPR'22 | 32.04 | 38.74 | 26.73 | 42.25 |
| SMILE+BiC Alssum et al. (2023) | ✓ | CVPR'23 | 52.24 | 6.25 | 48.22 | 0.31 |
| SMILE+iCaRL Alssum et al. (2023) | ✓ | CVPR'23 | 46.58 | 7.34 | 45.77 | 4.57 |
| CSTA (Vivit) Chen et al. (2025) | ✓ | TCSVT'25 | 54.98 | 5.06 | 51.01 | 6.91 |
| CSTA (Times) Chen et al. (2025) | ✓ | TCSVT'25 | 56.09 | 4.97 | 52.20 | 6.89 |
| **Ours** | ✗ | – | **57.83** | 14.01 | **53.95** | 15.09 |

Table 4: Ablation Study on UCF101 and HMDB51, with best results in bold.

| Idx | $\mathcal{A}^b$ | FSSD | TD-MoE | UCF101(5 × 10s) Acc ↑ | $\overline{\text{Acc}}$ ↑ | BWF ↓ | HMDB51(5 × 5s) Acc ↑ | $\overline{\text{Acc}}$ ↑ | BWF ↓ | UCF101(10 × 5s) Acc ↑ | $\overline{\text{Acc}}$ ↑ | BWF ↓ | HMDB51(25 × 1s) Acc ↑ | $\overline{\text{Acc}}$ ↑ | BWF ↓ |
|---|---|---|---|---|---|---|---|---|---|---|---|---|---|---|---|
| 1 | – | – | – | 70.65 | 74.67 | 7.01 | 43.30 | 43.62 | **6.18** | 70.14 | 72.72 | 5.33 | 43.71 | 47.48 | 8.74 |
| 2 | ✓ | – | – | 74.23 | 79.21 | 8.88 | 48.99 | 55.95 | 10.47 | 74.71 | 78.68 | 8.54 | 52.92 | 57.10 | 9.99 |
| 3 | ✓ | ✓ | – | 77.55 | 81.84 | 5.76 | 53.23 | 55.67 | 7.73 | 77.63 | 82.06 | **4.86** | 54.63 | 60.83 | 9.01 |
| 4 | – | – | ✓ | 79.33 | 89.36 | 12.38 | 56.12 | 61.14 | 11.75 | 85.94 | 93.47 | 7.40 | 62.54 | 68.88 | 10.20 |
| 5 | ✓ | – | ✓ | 83.07 | 91.28 | 10.52 | 57.47 | 63.37 | 21.30 | 88.03 | 94.14 | 8.39 | 64.71 | 73.02 | 21.72 |
| 6 | ✓ | ✓ | ✓ | **85.79** | **92.13** | **5.63** | **63.04** | **68.12** | 11.04 | **88.85** | **94.67** | 6.31 | **69.61** | **75.07** | **7.02** |

storing exemplars, surpasses all baselines across datasets and settings. On the TCD benchmark, our method outperforms the state-of-the-art approach (ST-Prompt†) as well as all exemplar-based methods on UCF101, HMDB51, and SSv2. On the vCLIMB benchmark, exemplar-based methods can alleviate forgetting by replaying stored samples,

which makes forgetting lower. Nevertheless, our method achieves higher final accuracy, surpassing the current state-of-the-art (CSTA) and all exemplar-based counterparts.

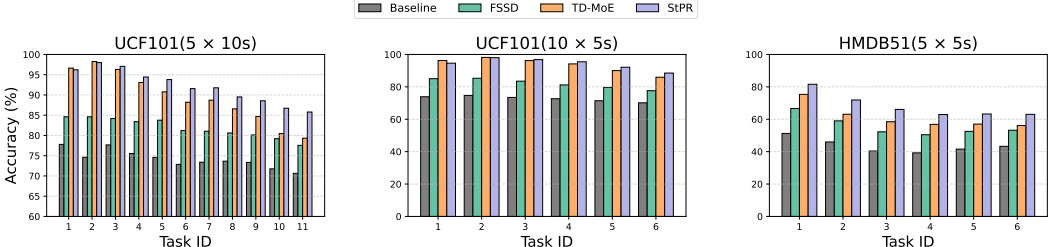

Figure 4: Task-wise ablation analysis across incremental tasks on UCF101 and HMDB51.

## 4.3 ABLATION STUDY

We perform ablation studies to evaluate the contribution of each component: the adapter tuning ($\mathcal{A}^b$), Frame-Shared Semantics Distillation (FSSD), and Temporal Decomposition-based Mixture-of-Experts (TD-MoE). Results are summarized in Table 4. The baseline (pretrained CLIP) model exhibits limited performance on downstream tasks, as it lacks adaptation to new task-specific categories. Introducing FSSD alone moderately improves performance by preserving spatial semantics and reducing semantic drift, while TD-MoE independently enhances adaptation by leveraging temporal dynamics. However, using either module alone yields suboptimal performance. Combining adapter tuning with TD-MoE provides further improvements but still lacks sufficient stability in preserving spatial semantics. The full model (StPR), integrating both FSSD and TD-MoE, achieves the most stable performance across tasks, demonstrating the complementary strengths of spatial semantic preservation and temporal dynamic modeling.

As shown in Figure 4, we progressively add our modules (FSSD and TD-MOE) on the UCF101 and HMDB51 datasets. Significant improvements are observed at each incremental stage, with more pronounced gains as the number of tasks increases, especially in the long-term scenario (10 tasks). This further validates the effectiveness of our proposed method and the superior performance of our model.

### 4.4 Further analysis

**Analysis of temporal-decomposition routing strategies.** Table 5 compares our TD-MoE with several alternative Mixture-of-Experts (MoE) Jacobs et al. (1991) routing strategies. Simple averaging (Avg-MoE) and static weight assignments—including CLIP-MoE, which uses frozen CLIP visual features for routing, and Adapter-MoE, which uses adapter-tuned CLIP features—provide moderate improvements but fail to dynamically leverage task-specific temporal cues, often resulting in higher forgetting. In contrast, TD-MoE enables adaptive expert weighting based on temporal dynamics, consistently improving both accuracy and stability across tasks. This highlights the importance of modeling temporal variability explicitly, rather than relying on static or feature-agnostic routing.

Table 5: MoE Method Results on UCF101 and HMDB51, with best results in bold.

| Method | UCF101($5 \times 10s$) | | HMDB51($5 \times 5s$) | | UCF101($10 \times 5s$) | | HMDB51($25 \times 1s$) | |
|---|---|---|---|---|---|---|---|---|
| | Acc ↑ | BWF ↓ | Acc ↑ | BWF ↓ | Acc ↑ | BWF ↓ | Acc ↑ | BWF ↓ |
| Avg-MoE | 81.14 | 9.95 | 59.46 | 8.49 | 84.04 | 9.60 | 62.69 | 9.14 |
| CLIP-MoE | 83.59 | 7.85 | 58.82 | 10.01 | 85.80 | 7.27 | 65.43 | 8.08 |
| Adapter-MoE | 83.26 | 6.07 | 61.99 | **7.75** | 84.31 | 7.82 | 65.57 | 7.68 |
| TD-MoE(Ours) | **85.79** | **5.63** | **63.04** | 11.04 | **88.52** | **6.39** | **69.61** | **7.02** |

Table 6: Distillation Method Results on UCF101 and HMDB51, with best results in bold.

| Method | UCF101($5 \times 10s$) | | HMDB51($5 \times 5s$) | | UCF101($10 \times 5s$) | | HMDB51($25 \times 1s$) | |
|---|---|---|---|---|---|---|---|---|
| | Acc ↑ | BWF ↓ | Acc ↑ | BWF ↓ | Acc ↑ | BWF ↓ | Acc ↑ | BWF ↓ |
| w/o Distillation | 83.07 | 10.52 | 57.47 | 21.30 | 88.03 | 8.39 | 64.71 | 21.72 |
| Distillation | 84.27 | 7.45 | 61.74 | 13.33 | 88.12 | 7.95 | 67.54 | 13.38 |
| FSSD(Ours) | **85.79** | **5.63** | **63.04** | **11.04** | **88.52** | **6.39** | **69.61** | **7.02** |

**Effectiveness of FSSD over Uniform Distillation.** Table 6 compares our FSSD method with the no-distillation baseline (w/o Distillation) and standard uniform distillation (Distillation) across four VCIL settings. While uniform distillation improves accuracy and reduces backward forgetting (BWF) over the naive baseline, FSSD consistently outperforms both, achieving the highest accuracy and lowest BWF in all settings. These results highlight the benefit of selectively preserving frame-shared semantics, validating the importance-aware design of FSSD for continual video learning. For more experiments, see the appendix C

## 5 Conclusion

In this work, we propose StPR, a unified and exemplar-free framework for Video Class-Incremental Learning (VCIL) to tackle the spatiotemporal challenges in continual video learning. By disentangling spatial semantics and temporal dynamics, StPR effectively balances stability and plasticity without relying on stored exemplars. Our method combines Frame-Shared Semantics Distillation (FSSD), which selectively preserves meaningful and stable semantic channels, protecting model's plasticity. Temporal-Decomposition-based Mixture-of-Experts (TD-MoE), adaptively routes inputs based on temporal cues, reducing forgetting in deep networks. Extensive experiments on UCF101, HMDB51, and SSV2 validate the effectiveness and efficiency of our approach, establishing new state-of-the-art results for continual video recognition. In future work, we plan to explore more realistic application scenarios, such as open-world settings, and investigate the deployment of our method on resource-constrained edge devices. See Appendix D for our statement on LLM usage.

## 6 ACKNOWLEDGMENTS

This work was supported by the New Generation Artificial Intelligence-National Science and Technology Major Project (2025ZD0123601), in part by the National Natural Science Foundation of China under Grants 62576262, U22A2096, in part by the Key Research and Development Program of Shaanxi Province under grant 2024GX-YBXM135, 2024SF-YBXM-647, in part by the Fundamental Research Funds for the Central Universities under Grant QTZX25083, QTZX23042.

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

# A  ALGORITHM

---

**Algorithm 1:** Frame-Shared Semantics Distillation (FSSD)

---

**Input:** Current task data $\mathcal{D}_b$; frozen model from task $b-1$; text features $\{\mathbf{T}_c\}$

**Output:** FSSD loss $\mathcal{L}_{\text{FSSD}}$

**for** *each class* $c \in \mathcal{Y}_b$ **do**

    Compute mean spatial features:

        $\bar{\mathbf{V}}_{b,c}^s = \frac{1}{N_f} \sum_{i=1}^{N_f} \mathbf{V}_{b,c,i}^s$ ;                            `// Current model`

        $\bar{\mathbf{V}}_{b-1,c}^s = \frac{1}{N_f} \sum_{i=1}^{N_f} \mathbf{V}_{b-1,c,i}^s$ ;                    `// Previous model`

    **for** *each channel* $j = 1$ *to* $d$ **do**

        Estimate $\mu_{c,j}, \sigma_{c,j}^2$ from $\mathbf{V}_{b-1,c}^s$ ;          `// Across frames`

        Compute semantic sensitivity:

            $\mathcal{I}(\mu_{c,j}) = \frac{1}{\sigma_{c,j}^2}$ ;                   `// Fisher Information`

        Compute classification contribution:

            $\mathbb{E}[\gamma_{c,j}] \propto \frac{T_{c,j} \cdot \mu_{c,j}}{\lambda}$ ;             `// Cosine-aligned score`

        Compute importance score:

            $I_{b-1,c,j} = \frac{T_{c,j} \cdot \mu_{c,j}}{\sigma_{c,j}^2}$ ;             `// Weighted relevance`

Compute weighted distillation loss:

$\mathcal{L}_{FSSD} = \frac{1}{|\mathcal{D}^b| \cdot d_{vt}} \sum_c^{|\mathcal{C}_b|} \sum_i^{N_c} \sum_j^{d_{vt}} I_{b-1,c,j} \cdot \|\bar{V}_{b-1,c,i,j}^s - \bar{V}_{b,c,i,j}^s\|_2^2$

**return** $\mathcal{L}_{FSSD}$

---

## A.1  ALGORITHM OF TD-MoE

---

**Algorithm 2:** Temporal Decomposition based Mixture-of-Experts Inference

---

**Input:** Video frames $\{\mathbf{x}_i^V\}_{i=1}^{N_f}$;

    Task-specific experts $\{\mathcal{G}_k\}_{k=1}^K$;

    Temporal anchors $\{\bar{\mathbf{V}}_c^{tem}\}_{c=1}^C$ for current task

**Output:** Final representation $\mathbf{V}$

Compute spatial mean: $\bar{\mathbf{V}}^s = \frac{1}{N_f} \sum_{i=1}^{N_f} \mathbf{x}_i^V$ ;      `// Mean of frame features`

**for** *each expert* $k = 1$ *to* $K$ **do**

    Concatenate CLS token and frame features

    $\mathbf{V}_k^{st} = \mathcal{G}_k([\mathbf{x}_{CLS}^V; \mathbf{x}_1^V; \ldots; \mathbf{x}_{N_f}^V])[0]$ ;     `// Spatiotemporal feature`

    Compute temporal representation:

    $\mathbf{V}_k^{tem} = \mathbf{V}_k^{st} - \bar{\mathbf{V}}^s$ ;              `// Temporal decomposition`

    Compute routing score:

    $r_k = \max_{c \in \mathcal{Y}_k} \cos(\mathbf{V}_k^{tem}, \bar{\mathbf{V}}_c^{tem})$ ;    `// Similarity to temporal anchors`

Compute final representation:

$\mathbf{V} = \bar{\mathbf{V}}^s + \sum_{k=1}^K r_k \cdot \mathbf{V}_k^{st}$ ;            `// Residual fusion`

**return** $\mathbf{V}$

---

**Frame-Shared Semantics Distillation (FSSD).** Algorithm 1 mitigates forgetting by selectively preserving spatial feature channels that are semantically important and temporally stable across frames. Importance is computed per channel using two criteria: (1) *Semantic sensitivity*, quantified by Fisher Information, and (2) *Classification contribution*, measured by cosine similarity with text features. These weights are used in a weighted distillation loss between the frozen previous model and the current task model, enabling exemplar-free knowledge retention while allowing plasticity.

**Temporal Decomposition-based Mixture-of-Experts (TD-MoE).** Algorithm 2 routes video inputs to task-specific experts based on temporal relevance. Each expert encodes spatiotemporal features from video frames, from which temporal dynamics are isolated via residual decomposition.

Temporal features are then compared to precomputed class anchors to compute routing scores. Final representations are generated by combining the expert outputs with the spatial feature via residual fusion. This enables dynamic, task ID-agnostic inference driven by temporal structure.

# B  THEORETICAL SUPPLEMENT

## B.1  FRAME-SHARED SEMANTICS

**Semantic Sensitivity.** The Fisher Information with respect to the mean parameter $\mu_j$ is defined as:

$$\mathcal{I}_j(\mu_{c,j}) = \mathbb{E}_{\bar{V}_{c,j}^s}\left[\left(\frac{\partial}{\partial\mu_{c,j}}\log p(\bar{V}_{c,j}^s;\mu_{c,j})\right)^2\right], \tag{16}$$

where $p(\bar{V}_{c,j}^s;\mu_{c,j})$ is the probability density function of the Gaussian:

$$p(\bar{V}_{c,j}^s;\mu_{c,j}) = \frac{1}{\sqrt{2\pi\sigma_{c,j}^2}}\exp\left(-\frac{(\bar{V}_{c,j}^s-\mu_{c,j})^2}{2\sigma_{c,j}^2}\right). \tag{17}$$

Taking the derivative of the log-likelihood with respect to $\mu_i$:

$$\log p(\bar{V}_{c,j}^s;\mu_{c,j}) = -\frac{1}{2}\log(2\pi\sigma_{c,j}^2) - \frac{(\bar{V}_{c,j}^s-\mu_{c,j})^2}{2\sigma_i^2}, \tag{18}$$

$$\frac{\partial}{\partial\mu_{c,j}}\log p(\bar{V}_{c,j}^s;\mu_{c,j}) = \frac{\bar{V}_{c,j}^s-\mu_{c,j}}{\sigma_{c,j}^2}. \tag{19}$$

Then, the Fisher Information becomes:

$$\mathcal{I}(\mu_{c,j}) = \mathbb{E}\left[\left(\frac{\bar{V}_{c,j}^s-\mu_{c,j}}{\sigma_{c.j}^2}\right)^2\right] = \frac{1}{\sigma_j^4}\mathbb{E}\left[(\bar{V}_{c,j}^s-\mu_{c,j})^2\right] = \frac{1}{\sigma_j^4}\cdot\sigma_{c,j}^2 = \frac{1}{\sigma_{c,j}^2}. \tag{20}$$

Thus, we obtain:

$$\mathcal{I}(\mu_{c,j}) = \frac{1}{\sigma_{c,j}^2}. \tag{21}$$

**Classification Contribution.** As we use cosine distance as classification basis, the $c$-th classification decision score $\gamma_{c,j}$ of the $j$-th channel is modeled as:

$$\gamma_{c,j} = \frac{\bar{V}_{c,j}^s\cdot T_{c,j}}{\|\bar{V}_c^s\|\cdot\|T_c\|} \propto \frac{\bar{V}_{c,j}^s\cdot T_{c,j}}{\|\bar{V}_c^s\|}. \tag{22}$$

Therefore, the score function s can be approximately written as:

$$\gamma_{c,j} = \sum_j \bar{V}_{c,j}^s\cdot\alpha_j, \quad \alpha_j = \frac{T_{c,j}}{\|\bar{V}_c^s\|} \approx \frac{T_{c,j}}{\lambda}. \tag{23}$$

For the expected score $\mathbb{E}[\gamma_{c,j}]$, it can be calculated as:

$$\mathbb{E}[\gamma_{c,j}] \propto \mathbb{E}\left[\frac{\sum_i \bar{V}_{c,j}^s\cdot T_{c,j}}{\|\bar{x}_v\|}\right] = \sum_j \alpha_j\cdot\mu_i \approx \frac{T_{c,j}\cdot\mu_{c,j}}{\lambda} \tag{24}$$

Then, the joint measure of informativeness is:

$$I_{c,j} \propto \alpha_j\cdot\mu_{c,j}\cdot\mathcal{I}(\mu_{c,j}) \approx \frac{T_{c,j}}{\lambda}\cdot\frac{\mu_{c,j}}{\sigma_{c,j}^2}. \tag{25}$$

This expression provides a theoretically principled and interpretable metric for frame-shared semantics. It reflects the intuition that an informative channel should (i) be strongly activated on average

($\mu_{c,j}$ large), and (ii) exhibit consistent activation patterns across samples ($\sigma^2_{c,j}$ small). Therefore, we define the importance of channel $j$ as:

$$I_{c,j} = \frac{T_{c,j} \cdot \mu_{c,j}}{\sigma^2_{c,j}}. \tag{26}$$

This formulation also aligns with the the signal-to-noise ratio theory (SNR), providing a unified theoretical justification.

## C  ADDITIONAL EXPERIMENTS

### C.1  ANALYSIS OF HYPER-PARAMETER.

Table 7: Analysis of hyperparameter $w$ on HMDB51 and UCF101 datasets. Best results are in bold.

| Dataset | Metric | $1 \times 10^3$ | $1 \times 10^4$ | $2.5 \times 10^4$ | $5 \times 10^4$ | $1 \times 10^5$ |
|---|---|---|---|---|---|---|
| HMDB($5 \times 5$s) | Acc ↑ | 59.51 | 63.04 | 62.10 | 62.29 | **63.05** |
| | BWF ↓ | 16.90 | **11.04** | 11.14 | 12.32 | 12.77 |
| UCF101($5 \times 10$s) | Acc ↑ | 85.10 | **85.79** | 84.54 | 84.88 | 84.16 |
| | BWF ↓ | 7.93 | **5.63** | 6.95 | 5.67 | 6.83 |

Table. 7 analyzes the sensitivity of the hyper-parameter $w$, which controls the strength of FSSD regularization. Across a wide range of values, our framework maintains stable performance, indicating robustness to hyper-parameter variations. Moderate $w$ values achieve the best trade-off between knowledge retention and adaptability, avoiding under-regularization or excessive constraint on model plasticity.

### C.2  COMPARISON OF MoE ROUTING STRATEGIES (TASK BY TASK)

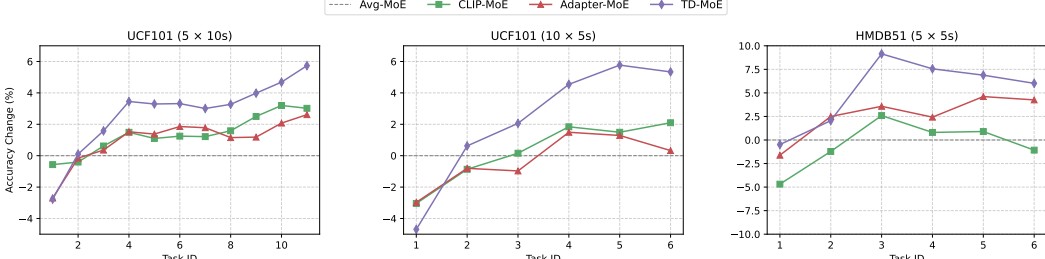

Figure 5: MoE Methods Performance per Task on UCF101 and HMDB51

Figure 5 presents the relative accuracy change (%) of different Mixture-of-Experts (MoE) routing strategies on three benchmarks: UCF101 with two task configurations (5×10s and 10×5s), and HMDB51 (5×5s). The horizontal axis represents the incremental task ID, while the vertical axis shows the accuracy change relative to the Avg-MoE baseline.

We observe that **TD-MoE consistently outperforms** all baselines across datasets and task granularities. Its performance advantage becomes more pronounced as the number of tasks increases, reaching up to 6–8% improvement on later tasks. In contrast, **Adapter-MoE and CLIP-MoE** exhibit only marginal gains, which tend to saturate early, suggesting limited ability to model task-specific dynamics. These findings confirm that temporal decomposition is effective for guiding expert selection in a task ID-agnostic manner, and helps mitigate forgetting by capturing relevant spatiotemporal cues.

As shown in 6, temporal decomposition-based router is effective at task boundary decisions.

As shown in Figure 4, it maintains highest accuracy over time and effectively mitigates forgetting, demonstrating the complementary strengths of spatial semantic preservation and temporal dynamic modeling.

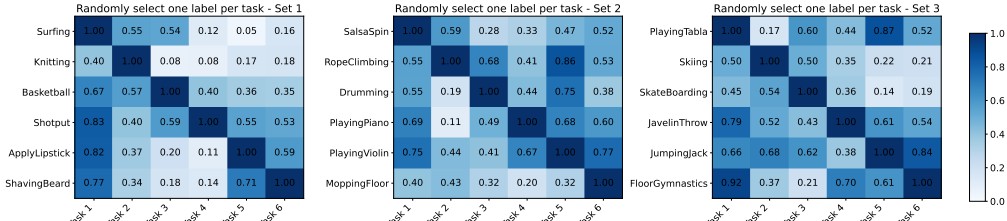

Figure 6: Heatmap of task selection by temporal decomposition-based router on the UCF101.

To investigate the performance of TD-MoE on temporally challenging datasets, we added an ablation study on SSv2 with and without the TD-MoE module(Table 8). Since SSv2 is highly sensitive to temporal cues, TD-MoE brings a clear performance improvement, which aligns with our design motivation. Without TD-MoE, the model relies on a shared MSA module in the deeper layers. Because the dataset itself is difficult to train and each new task initializes the MSA module with the parameters from the previous task, the model's ability to learn new tasks is inherently weakened. This leads to poor performance on new tasks; thus, as more tasks are added, forgetting does not further deteriorate simply because the initial performance is already low. Nevertheless, the shared deep MSA module still experiences substantial forgetting. In contrast, with TD-MoE, the model learns temporal patterns more effectively, achieving higher accuracy on new tasks while not exhibiting a noticeable increase in forgetting as the number of tasks grows.

Table 8: SSv2 Performance Across Task Numbers (w/ vs. w/o TD-MoE). Best results are in bold.

| Task Numbers | TD-MoE | $\overline{\text{Acc}}\uparrow$ | BWF↓ |
|---|---|---|---|
| 10s | ✗ | 17.64 | 56.32 |
| 10s | ✓ | **40.79** | **18.86** |
| 20s | ✗ | 18.39 | 49.49 |
| 20s | ✓ | **37.30** | **19.44** |

## C.3 EFFECTIVENESS OF FRAME-SHARED SEMANTICS DISTILLATION

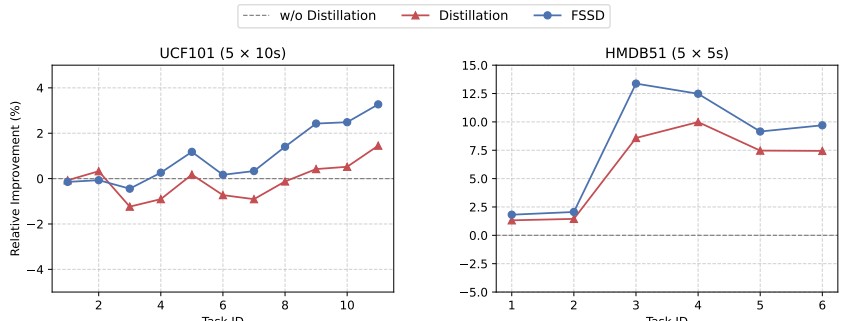

Figure 7: Distillation Methods Performance per Task on UCF101 and HMDB51

Figure 7 compares the impact of different distillation strategies, including w/o Distillation, unified Distillation, and our proposed Frame-Shared Semantics Distillation (FSSD), on UCF101 and HMDB51. The vertical axis reports the relative improvement over the non-distillation baseline.

The results demonstrate that **FSSD delivers the most consistent and significant improvements**, particularly on HMDB51. While unified distillation offers slight improvements, it lacks consistency across tasks. This suggests that uniform constraints fail to address the heterogeneous semantic importance of feature channels. Moreover, the increasing gap between FSSD and other methods over time confirms that **adaptive regularization based on frame-shared semantic importance is critical** for preserving relevant knowledge across tasks in VCIL.

To compare with gradient-based regularization methods and clarify the difference between our approach and EWC Kirkpatrick et al. (2017), we provide the following explanation and comparison experiments. Gradient-based methods such as EWC rely on parameter-level regularization. We reproduced EWC and compared it with FSSD (Table 9). On both UCF101 and HMDB51, our semantic-importance–based approach consistently outperforms EWC. This is because EWC approximates the Fisher information using only diagonal elements, ignoring cross-parameter correlations and assuming model stability during estimation, which introduces approximation error. Moreover, EWC does not accumulate importance across tasks; it assumes that the model after fine-tuning already preserves previous knowledge. As the number of tasks grows, feature drift becomes more significant, causing the Fisher-based estimate to become increasingly inaccurate. In contrast, FSSD computes importance via an analytic, feature-semantic formulation and accumulates knowledge over sessions, allowing it to better preserve early-task information and achieve superior performance. In contrast, FSSD computes importance analytically via feature statistics (mean and variance) aligned with the textual representations, requiring no backpropagation and thus incurring negligible computational cost. Furthermore, the importance vector in FSSD has the same dimensionality as the output features, resulting in minimal memory footprint. EWC, however, must compute gradients for all samples and store importance values for all trainable parameters, leading to significantly higher computational and memory overhead.

Table 9: Performance of EWC and our method on UCF101 and HMDB51.

|  | Methods | $\overline{\text{Acc}}$ ↑ | Acc ↑ |
|---|---|---|---|
| UCF101 (10×5s) | EWC Kirkpatrick et al. (2017) | 93.49 | 86.37 |
| UCF101 (10×5s) | FSSD | **94.67** | **88.85** |
| UCF101 (5×10s) | EWC Kirkpatrick et al. (2017) | 90.00 | 80.29 |
| UCF101 (5×10s) | FSSD | **92.13** | **85.79** |
| HMDB51 (1×25s) | EWC Kirkpatrick et al. (2017) | 66.49 | 55.60 |
| HMDB51 (1×25s) | FSSD | **67.01** | **57.89** |

## C.4 Parameter Count Analysis

We compute the parameter count each module: the CLIP ViT-B/16 backbone, the inserted adapters, and the spatiotemporal encoder module.

Table 10: Parameter counts for each module per 8-frame video.

| Module | Parameters |
|---|---|
| CLIP (ViT-B/16) (frozen) | 86M |
| Adapter | 1.17M (1.36%) |
| Spatiotemporal Encoder | 9.45M (10.99%) |

Table 10 provides a detailed breakdown of parameter count for each component in our framework, evaluated on 8-frame video inputs. The backbone CLIP (ViT-B/16) dominates the overall cost with 86M parameters. The inserted adapter modules, despite being integrated into every transformer layer, introduce only 1.17M additional parameters (1.36%), demonstrating their lightweight nature. Furthermore, our spatiotemporal encoder used to capture dynamic information, adds 9.45M parameters (10.99%).

Table 11: Performance of TD-MoE with Different Frame Counts on UCF101 (10×5s).

| Frames | $\overline{\text{Acc}}$ ↑ | Acc ↑ | BWF ↓ |
|---|---|---|---|
| 2 | 91.29 | 83.75 | 8.91 |
| 4 | 92.72 | 85.29 | 8.60 |
| 8 | 94.67 | 88.85 | 6.31 |
| 16 | 94.50 | 89.54 | 5.58 |
| 32 | 93.39 | 89.48 | 4.72 |

To analyze TD-MoE's ability to handle different degrees of temporal information, we conducted additional experiments on UCF101 with input clips containing sparse (2, 4 frames), adequate (8, 16 frames), and redundant (32 frames) temporal cues. The results are shown in Table 11. We observe that performance increases as more temporal information becomes available, peaking at 8–16 frames. Notably, even when the frame count is doubled to 32, the accuracy remains stable, indicating that TD-MoE can effectively exploit longer temporal sequences without being negatively affected by redundancy. These results confirm that TD-MoE maintains strong long-term temporal modeling capability.

Table 12: Performance of different layers on UCF101 (10×5s).

|  | $\overline{\text{Acc}}$ ↑ | Acc ↑ | Parameters ↓ |
|---|---|---|---|
| CLIP (backbone) | 74.67 | 70.65 | 149M |
| 1 layer | 91.29 | 83.75 | 3M (2.01%) |
| 2 layers | 92.72 | 85.29 | 6M (4.02%) |
| 3 layers | 94.67 | 88.85 | 9M (6.03%) |

Table 13: Performance of selecting latest 5 experts on UCF101.

|  | Select latest 5 | $\overline{\text{Acc}}$ ↑ | Acc ↑ | Linear Growth |
|---|---|---|---|---|
| UCF101 (5×10s) | ✓ | 88.77 | 74.97 | ✗ |
| UCF101 (5×10s) | ✗ | 92.13 | 85.79 | ✓ |
| UCF101 (2×25s) | ✓ | 82.15 | 71.10 | ✗ |
| UCF101 (2×25s) | ✗ | 88.52 | 82.67 | ✓ |

We conducted preliminary studies on expert design. As shown in Table 12, reducing expert depth lowers parameters and slightly affects temporal modeling, but even a single-layer expert with TD-MoE still achieves state-of-the-art performance ($\overline{\text{Acc}} = 84.75$). We further evaluate a efficient design: we tested a simple selective-expert strategy by keeping only latest five experts Table 13. Because the inputs of all experts and temporal anchors rely on shared spatial features, newly trained experts naturally retain part of the old knowledge. Under this mechanism, the 2×25 configuration falls slightly short of the current state-of-the-art ($\overline{\text{Acc}} = 85.67$), while the 5×10 setting still exceeds it ($\overline{\text{Acc}} = 84.75$). This demonstrates that selective expert retention is feasible. In future work, we plan to further explore more principled expert-selection mechanisms and investigate how to transfer knowledge across experts to reduce the performance gap when using a fixed number of experts.

# D  LLM USAGE STATEMENT

In accordance with the ICLR 2026 policy on large language models (LLMs), we clarify that LLMs were employed solely to assist in polishing the language and improving readability of the manuscript. The conception of the research problem, development of the methodology, algorithmic design, code implementation, experimental setup, and result analysis were entirely carried out by the authors without reliance on LLMs.

