# OpenReview forum: "StPR: Spatiotemporal Preservation and Routing for Exemplar-Free Video Class-Incremental Learning"
_ICLR.cc/2026/Conference — ICLR 2026 Poster_

### Official Review · Reviewer_s53Z · 2025-10-27

**Soundness:** 3
**Presentation:** 2
**Contribution:** 2
**Rating:** 4
**Confidence:** 3

**Summary:**

The topic of this paper is about video class-incremental learning(VCIL). The authors propose an exemplar-free VCIL framework with the spatio-temporal Preservation Routing(STPR) mechanism. It consists of a frame-shared semantics distillation(FSSD) and a Temporal Decomposition-based Mixture-of-Experts(TD-MoE) strategy. The STPR framework has been evaluated on several datasets.

**Strengths:**

+ The idea of exploring temporal-based MoE in VCIL is interesting.
+ The performance of the proposed method StPR outperfoms that of other methods by a margin on several datasets.

**Weaknesses:**

- Concerns about the FSSD module. The proposed FSSD strategy is a plug-in method can be integrated to other VCIL methods. To show the effectivess of FSSD, can it be evaluated on different VCIL baselines? The authors have conducted ablation studies in Table 4 and the performance with only employing the FSSD method is needed.

- The presentation can be improved. For example, there are some minor spelling mistakes. “suppl.”->supplemental. The formulation “$A^b$” is not presented in Figure 2 of the overview framework.

- Lack of some important comparisons. The idea of decoupling spatiotemporal knowledge in VCIL has been discussed in previous methods (e.g., [a][b] ). Compare to these works and make a discussion in the related work. Besides, some works published in 2025(e.g., [b]-[c]) can be compared and discussed. For instance, the proposed method can utilize the complex large-scale video continual learning benchamrks [c] for evaluation.

[a] When Video Classification Meets Incremental Classes, MM2021

[b] Learning Conditional Space-Time Prompt Distributions for Video Class-Incremental Learning, CVPR 2025

[c] CRAM:Large-scale Video Continual Learning with Bootstrapped Compression, ICCV 2025

**Questions:**

My major concerns are inclued in the above weaknesses.

Q1. Different parameter-efficient finetuning(PEFT) strategies(e.g., prompt, adapter, MoE) have been utilized in VCIL scenarios. How to select appropriate PEFT strategy in different practical VCIL scenarios?

---

> ### Author Response · Authors · 2025-11-22
> **Response to Reviewer s53Z: Part 1 (W1, W2, Q1)**
>
> Thank you for your guidance and suggestions. They are very helpful for improving our work. We address your questions point by point below. We address each point in detail below. The additional experiments and discussions will be updated in the camera-ready version.
>
> **W1:**
> To demonstrate the plug-and-play nature of our method, we select the open-source work TCD [1], reproduce it as our baseline, and add our FSSD method on top of it (**Table 1**). CNN is a standard classification protocol, where the model classifies the data using the trained fully-connected layer. NME, which compares the feature representation of test data with the mean-of-exemplars.
> Moreover, our ablation study (**Table 2**) includes an experiment using FSSD alone, where we take the Adapter Tuning model as the baseline and then add our FSSD module, showing consistent performance gains when the module is added, further validating its effectiveness.
>
> **Table 1: Ablation of FSSD on UCF101 (10 × 5s ) under TCD baseline.**
>
> |                   | NME: $\overline{\mathrm{Acc}} \uparrow$        |NME: $ Acc \uparrow$       | NME: BWF $\downarrow$ | CNN: $\overline{\mathrm{Acc}} \uparrow$        |CNN: $ Acc \uparrow$       | CNN: BWF $\downarrow$ |
> |-------------------|------------------------------------------------|---------------------------|-----------------------|------------------------------------------------|---------------------------|-----------------------|
> | TCD [1]           |         75.65                                  |          71.00            |         3.73          |         72.46                                  |          65.90            |         7.04          |
> | + FSSD            |         **76.13**                              |          **72.38**        |         **3.15**      |         **72.92**                              |          **66.95**        |         **6.90**      |
>
> **Table 2: Ablation of FSSD on UCF101 and HMDB51.**
> |                  | UCF101 ($5 \times 10$s)|UCF101 ($5 \times 10$s)|UCF101 ($5 \times 10$s)|HMDB51 ($5 \times 5$s)|HMDB51 ($5 \times 5$s)|HMDB51 ($5 \times 5$s)| UCF101 ($10 \times 5$s)| UCF101 ($10 \times 5$s)| UCF101 ($10 \times 5$s)| HMDB51 ($25 \times 1$s)| HMDB51 ($25 \times 1$s)| HMDB51 ($25 \times 1$s)|
> |------------------|------------------------|-----------------------|------------------------|------------------------|------------------------|-----------------------|------------------------|------------------------|------------------------|-----------------------|------------------------|------------------------|
> |                  | $ Acc \uparrow$ | $\overline{\mathrm{Acc}} \uparrow$ |  BWF $\downarrow$ | $ Acc \uparrow$ | $\overline{\mathrm{Acc}} \uparrow$ |  BWF $\downarrow$ | $ Acc \uparrow$ | $\overline{\mathrm{Acc}} \uparrow$ |  BWF $\downarrow$ | $ Acc \uparrow$ | $\overline{\mathrm{Acc}} \uparrow$ |  BWF $\downarrow$ |
> | Adapter Tuning   |         74.23          |         79.21         |           8.88        |           48.99       |         55.95          |         10.47         |           74.71        |           78.68        |         8.54          |         52.92         |          57.10        |           9.99       |
> | +FSSD            |         **77.55**          |         **81.84**         |           **5.76**        |           **53.23**       |         **55.67**          |         **7.73**         |           **77.63**        |           **82.06**        |         **4.86**          |         **54.63**         |          **60.83**        |           **9.01**       |
>
> [1] Park J, Kang M, Han B. Class-incremental learning for action recognition in videos[C]//Proceedings of the IEEE/CVF international conference on computer vision. 2021: 13698-13707.
>
> **W2:**
> Thank you for pointing out these issues. We have corrected the spelling mistakes and the corrections will be reflected in the camera-ready version.
>
> **Q1:**
> Regarding the selection of PEFT, here is our explanation:
>
> We follow the findings of SSIAT [2], which show that under Transformer-based architectures, adapter-based finetuning consistently outperforms prompt-based finetuning in CIL settings through extensive experiments. Based on this empirical evidence, we adopt adapter tuning as the PEFT strategy in our framework.
>
> [2]. Tan Y, Zhou Q, Xiang X, et al. Semantically-shifted incremental adapter-tuning is a continual vitransformer[C]//Proceedings of the IEEE/CVF Conference on Computer Vision and Pattern Recognition. 2024: 23252-23262.

---

> ### Author Response · Authors · 2025-11-22
> **Response to Reviewer s53Z: Part 2 (W3)**
>
> **W3:**
> We thank the reviewer for pointing out these related directions and will explicitly discuss them in the revised related work and experiments. Below is our discussion and comparison:
>
> SM [3] introduces the task of Class-Incremental Video Classification (CIVC) and proposes a framework using spatio-temporal decomposed distillation and dual granularity exemplar selection to address catastrophic forgetting. The method is innovative and practical, achieving significant accuracy improvements with reduced memory usage. CoSTEP [4] constructs a “frame grid’’ and applies patch-wise attention followed by a diffusion-based U-Net to implicitly model spatiotemporal relations. While effective, its diffusion backbone is computationally heavy and used to maintain old knowledge via a generative process. In contrast, our TD-MoE explicitly formulates temporal residuals and inter-frame information and uses these disentangled temporal anchors to route experts, leading to a lightweight, fully exemplar-free mechanism that directly targets temporal structure.
>
> We also compare against these methods. To ensure a fair comparison and demonstrate the performance of our algorithm, for SM, we test on the same subset of SSv2 with an equal number of classes while maintaining the exemplar-free setting. As shown in **Table 3**, even in the exemplar-free setting, our method still achieves significant improvements compared to SM, which is exemplar-based method, proving the effectiveness of our approach. **Table 4** shows that StPR outperforms CoSTEP on all datasets except SSv2 with 9 tasks; when the number of tasks increases to 18, StPR surpasses CoSTEP, suggesting better long-term robustness. CRAM [5] reduces replay cost via compressed feature codes but still requires large buffers (e.g., K-700 (1404Mb) and EK-100 (1464Mb)). In contrast, our method is evaluated on the large-scale Kinetics-400 dataset with 20 tasks and no replay (zero memory), and achieves a BWF of 15.09, which is lower than the 15.20 BWF reported by CRAM with replay over 10 tasks. These comparisons will be added and discussed, and they collectively support that our disentangled temporal modeling mitigates forgetting effectively without relying on replay, even on complex large-scale VCIL benchmarks. We will include this discussion in the related work and experiments of camera-ready version.
>
> **Table 3: Performance of SM and Our StPR Method on the SSv2 Subset.**
> |       Methods   | Venue  |     $ Acc \uparrow$  | BWF $\downarrow$ | Memory |
> |-----------------|--------|----------------------|------------------|--------|
> |      ST [3]     | MM2021 |          39.73       |       34.75      |  0.44G |
> |      SM [3]     | MM2021 |          45.84       |       32.25      |  0.61G |
> |    StPR (ours)  |        |       **49.71**      |  **15.55**       |   **0**|
>
> **Table 4: Performance of CoSTEP and Our StPR Method on UCF101, HMDB51 and SSv2.**
> |             |        |  UCF101 |  UCF101 |  UCF101 | HMDB51    | HMDB51    |  SSv2   |    SSv2   |
> |-------------|--------|---------|---------|---------| ----------|-----------|---------|-----------|
> |             | Venue  | 10 × 5s | 5 × 10s | 2 × 25s |  5 × 5s   |  1 × 25s  | 10 × 9s |  5 × 18s  |
> | CoSTEP [4]  |CVPR2025| 86.05   | 86.71   | 86.95   |  61.70    |  61.84    |**41.44**|   36.60   |
> | StPR (ours) |        |**94.67**|**92.13**|**88.52**| **68.12** | **67.01** | 40.79   | **37.30** |
>
> We once again thank you for your valuable comments and suggestions. We hope our responses have addressed your concerns and would be happy to clarify any remaining questions; we would greatly appreciate it if you could consider raising the score in light of these responses.
>
> [3]. Zhao H, Qin X, Su S, et al. When video classification meets incremental classes[C]//Proceedings of the 29th ACM International Conference on Multimedia. 2021: 880-889.
>
> [4]. Zou X, Ma W, Zhao S. Learning Conditional Space-Time Prompt Distributions for Video Class-Incremental Learning[C]//Proceedings of the Computer Vision and Pattern Recognition Conference. 2025: 4862-4873.
>
> [5]. Mall S, Henriques J F. CRAM: Large Scale Video Continual Learning with Bootstrapped Compression[C]//Proceedings of the IEEE/CVF International Conference on Computer Vision. 2025: 15045-15055.

---

### Official Review · Reviewer_SftT · 2025-10-29

**Soundness:** 3
**Presentation:** 3
**Contribution:** 3
**Rating:** 6
**Confidence:** 4

**Summary:**

This paper introduces Spatiotemporal Preservation and Routing (StPR), a novel mechanism designed to tackle the challenging problem of Video Class-Incremental Learning (VCIL) in an exemplar-free (memory-free) setting. The proposed method explicitly disentangles and preserves critical spatial and temporal representations, effectively mitigating catastrophic forgetting while maintaining adaptability. StPR consists of two key components: Frame-Shared Semantics Distillation (FSSD) and the Temporal Decomposition-based Mixture of Experts (TD-MoE). FSSD identifies and preserves semantically stable and meaningful spatial channels by jointly considering channel-wise sensitivity and classification contribution. TD-MoE dynamically routes task-specific spatiotemporal experts based on decomposed temporal dynamics, enabling adaptive learning without task identifiers or stored exemplars. Experimental results demonstrate that this memory-free approach achieves superior performance compared to rehearsal-based baselines such as BiC and iCaRL.

**Strengths:**

1. This paper introduces an effective rehearsal free method that can obtain state-of-the-art performance in different benchmarks like vCLIMB and TCD, outperforming both exemplar-based (e.g., FrameMaker, HCE) and exemplar-free (e.g., STSP, ST-Prompt) methods.
2. Temporal Decomposition-based Mixture-of-Experts (TD-MoE) provides a routing based on temporal residuals and attention-weighted deviations which allows the inference without task IDs.
3. They provide a well-designed distillation loss FSSD that leverages the Fisher information and classification contribution to compute per-channel relevance which help preserving model stability-plasticity.
4. The paper provides extensive evaluations across multiple benchmarks and metrics, along with a thorough ablation study isolating the contributions of each component.

**Weaknesses:**

1. The proposed TD-MoE instantiates a separate expert for each new task. Consequently, both memory and computation costs are expected to scale linearly with the number of tasks. The paper does not provide an explicit analysis or discussion on this scalability issue, nor on potential mitigation strategies.
2. It would be valuable to understand the limitations of the TD-MoE in temporal challenging datasets like SSv2. For instances, analizing the forgetting of the model with and without the TD-MoE module when the number of tasks increase.

**Questions:**

1. How does memory and compute cost scale with the number of tasks? Do you freeze old experts or allow fine-tuning?
2. How does FSSD compare to gradient-based importance as a measure of stability?
3. Can you discuss the different between FSSD and the pricipales of EWC, which also use the Fisher information?

---

> ### Author Response · Authors · 2025-11-22
> **Response to Reviewer SftT: Part 1 (Q1, W1)**
>
> Thank you very much for your support of our work. Your suggestions are highly constructive. We address each of your questions point-by-point below. We address each point in detail below. The additional experiments and discussions will be updated in the camera-ready version.
>
> **W1 \& Q1:**
> Regarding the growth of expert model parameters and the exploration of mitigating this issue, we conducted the following experiments and provide the corresponding explanation:
>
> Indeed, the growth of experts with the number of tasks is a common limitation of MoE-based task-specific design continual learning [1]. Since our motivation is to leverage temporal information for routing to mitigate forgetting, **our main contribution lies in disentangling temporal information for expert routing**. However, we still explore some preliminary studies on the design of experts. As shown in **Table 1** (percentages denote module–to–CLIP FLOPs/parameters ratios), reducing expert depth lowers parameters and slightly affects temporal modeling, but even a single-layer expert with TD-MoE still outperforms state-of-the-art performance ($\overline{\mathrm{Acc}} = 84.75$).
>
> To minimize the number of experts as much as possible, we further evaluate an efficient design: we tested a simple selective-expert strategy by keeping only latest five experts **Table 2** (percentages denote module–to–CLIP FLOPs/parameters ratios). Because the inputs of all experts and temporal anchors rely on shared spatial features, newly trained experts naturally retain part of the old knowledge. Under this selection mechanism, the standard incremental setting (5×10) still exceeds the SOTA performance ($\overline{\mathrm{Acc}} = 84.75$). Since we freeze the experts for old tasks, the trainable expert parameters remain fixed, which does not increase GPU memory usage. In addition, our computational cost is negligible compared to that of the CLIP backbone. This demonstrates that selective expert retention is feasible.
>
> In future work, we plan to further explore more principled expert-selection mechanisms and investigate how to transfer knowledge across experts to reduce the performance gap when using a fixed number of experts.
>
> **Table 1: Performance of different layers on UCF101 (10×5s).**
>
> |               | $\overline{\mathrm{Acc}} \uparrow$ | Acc $\uparrow$ | Parameters |Flops (GFLOPs)|
> |---------------|------------------------------------|----------------|------------|--------------|
> | 1 layer       |         91.29                      |         83.75  |  3M(2.01\%)|0.0214(0.01%) |
> | 2 layers      |         92.72                      |         85.29  |  6M(4.02\%)|0.0427(0.02%) |
> | 3 layers      |         94.67                      |         88.85  |  9M(6.03\%)|0.0854(0.03%) |
>
>
> **Table 2: Performance of selecting latest 5 experts.**
>
> |                |  Select latest 5 | $\overline{\mathrm{Acc}} \uparrow$ | Trainable Parameters |Full Flops (GFLOPs)   |
> |----------------|------------------|------------------------------------|----------------------|----------------------|
> | UCF101 (5×10s) |      √           |                88.77               |  9M   (6.03\%)       |      0.854 (0.32%)   |
> | UCF101 (5×10s) |      ×           |                92.13               |  9M   (6.03\%)       |      0.427 (0.16%)   |
> | UCF101 (2×25s) |      √           |                82.15               |  9M   (6.03\%)       |      2.135 (0.79%)   |
> | UCF101 (2×25s) |      ×           |                88.52               |  9M   (6.03\%)       |      0.427 (0.16%)   |
>
> [1]. Wang Y, Zhou D W, Ye H J. Integrating Task-Specific and Universal Adapters for Pre-Trained Model-based Class-Incremental Learning[C]//Proceedings of the IEEE/CVF International Conference on Computer Vision. 2025: 806-816.

---

> ### Author Response · Authors · 2025-11-22
> **Response to Reviewer SftT: Part 2 (Q2, Q3, W2)**
>
> **W2:**
> To investigate the performance of TD-MoE on temporally challenging datasets, we added an ablation study on SSv2 with and without the TD-MoE module (**Table 3**). Since SSv2 is highly sensitive to temporal cues, TD-MoE brings a clear performance improvement, which aligns with our design motivation. Without TD-MoE, the model relies on a shared MSA module in the deeper layers. Because the dataset itself is difficult to train and each new task initializes the MSA module with the parameters from the previous task, the model’s ability to learn new tasks is inherently weakened. This leads to poor performance on new tasks; thus, as more tasks are added, forgetting does not further deteriorate simply because the initial performance is already low. Nevertheless, the shared deep MSA module still experiences substantial forgetting. In contrast, with TD-MoE, the model learns temporal patterns more effectively, achieving higher accuracy on new tasks while not exhibiting a noticeable increase in forgetting as the number of tasks grows.
>
> **Table 3: SSv2 Performance Across Task Numbers (w/ vs. w/o TD-MoE).**
> | Task Numbers   |  TD-MoE   | $\overline{\mathrm{Acc}} \uparrow$ | BWF $\downarrow$ |
> |----------------|-----------|------------------------------------|------------------|
> | 10s            |    ×      |           17.64                    |        56.32     |
> | 10s            |    √      |           40.79                    |        18.86     |
> | 20s            |    ×      |           18.39                    |        49.49     |
> | 20s            |    √      |           37.30                    |        19.44     |
>
> **Q2 \& Q3:**
> To compare with gradient-based regularization methods and clarify the difference between our approach and EWC, we provide the following explanation and comparison experiments:
>
> Gradient-based methods such as EWC rely on parameter-level regularization, whereas our FSSD module performs feature distillation based on channel-wise semantic importance, making the two fundamentally different in their regularization mechanisms. To further compare their performance, we reproduced EWC and compared it with FSSD (**Table 4**). On both UCF101 and HMDB51, our semantic-importance–based approach consistently outperforms EWC.
>
> This is because EWC approximates the Fisher information using only diagonal elements, ignoring cross-parameter correlations and assuming model stability during estimation, which introduces approximation error. Moreover, EWC does not accumulate importance across tasks; it assumes that the model after fine-tuning already preserves previous knowledge. As the number of tasks grows, feature drift becomes more significant, causing the Fisher-based estimate to become increasingly inaccurate. In contrast, FSSD computes importance via an analytic, feature-semantic formulation and accumulates knowledge over sessions, allowing it to better preserve early-task information and achieve superior performance.
>
> In terms of computational efficiency, EWC must compute gradients for all samples and store importance values for all trainable parameters (e.g., the 12-layer adapter module embedded in the backbone, with **$1.18$M** trainable parameters), resulting in substantially higher computational and memory overhead. In contrast, our FSSD computes importance analytically using feature statistics (mean and variance) aligned with textual representations, requiring no backpropagation and thus incurring negligible computational cost. Moreover, the importance vector produced by FSSD has the same dimensionality as the output features (**$5.12\times 10^{-4}$M**), leading to a minimal memory footprint.
>
> **Table 4: Performance of EWC and our method on UCF101 and HMDB51.**
> |                |    Methods       | $\overline{\mathrm{Acc}} \uparrow$ | Acc $\uparrow$ |
> |----------------|------------------|------------------------------------|----------------|
> | UCF101 (10×5s) |      EWC         |                93.49               |      86.37     |
> | UCF101 (10×5s) |      FSSD        |                **94.67**           |      **88.85** |
> | UCF101 (5×10s) |      EWC         |                90.00               |      80.29     |
> | UCF101 (5×10s) |      FSSD        |                **92.13**           |      **85.79** |
> | HMDB51 (1×25s) |      EWC         |                66.49               |      55.60     |
> | HMDB51 (1×25s) |      FSSD        |                **67.01**           |      **57.89** |
>
> We once again thank you for your valuable comments and suggestions. We hope our responses have addressed your concerns and would be happy to clarify any remaining questions; we would greatly appreciate it if you could consider raising the score in light of these responses.

---

### Official Review · Reviewer_VZo2 · 2025-10-29

**Soundness:** 4
**Presentation:** 4
**Contribution:** 4
**Rating:** 6
**Confidence:** 4

**Summary:**

This paper proposes StPR (Spatiotemporal Preservation and Routing), a unified, exemplar-free framework for Video Class-Incremental Learning (VCIL). The key challenge in VCIL is mitigating catastrophic forgetting while preserving both frame-shared spatial semantics and temporal dynamics across incrementally arriving action classes—without storing past video exemplars due to memory/privacy constraints.

To address this, the authors introduce two core components:

Frame-Shared Semantics Distillation (FSSD): A channel-wise importance-aware distillation mechanism that preserves semantically stable and classification-relevant spatial features by combining semantic sensitivity (via Fisher Information) and classification contribution (via cosine alignment with CLIP text embeddings).

Temporal Decomposition-based Mixture-of-Experts (TD-MoE): A task-agnostic routing mechanism that decomposes spatiotemporal features into static (mean) and dynamic (residual) components. At inference, it routes inputs to task-specific temporal experts based solely on temporal dynamics, eliminating the need for task IDs or stored exemplars.

The framework is built on a frozen CLIP ViT backbone with lightweight adapters and a small spatiotemporal encoder per task. Experiments on UCF101, HMDB51, SSv2, and Kinetics400 under standard VCIL benchmarks (TCD and vCLIMB) show state-of-the-art performance, outperforming both exemplar-based and exemplar-free baselines. Ablation studies and analyses support the design choices.

**Strengths:**

Strengths

Originality: The disentanglement of frame-shared semantics and temporal dynamics for VCIL is novel. While MoE and distillation exist in CIL, their adaptation to video via temporal decomposition and semantic importance weighting is creative and domain-aware.

Quality: Rigorous experiments with strong baselines, multiple datasets, and thorough ablations. The use of CLIP aligns with modern vision-language trends.

Clarity: The framework is easy to follow (Figure 2), and algorithms are provided (Appendix A). The writing is concise and technical depth is balanced well.

Significance: Addresses a critical gap in VCIL—exemplar-free learning with temporal modeling. The method is efficient (low FLOPs), scalable, and applicable to real-world scenarios like edge devices (mentioned in conclusion).

**Weaknesses:**

Task-Specific Expert Scaling: TD-MoE allocates one spatiotemporal encoder per task. While efficient per expert (~9M params), this leads to linear growth in parameters with the number of tasks (e.g., 10 tasks → ~90M extra params). This may limit scalability in long-task sequences. The paper does not discuss parameter efficiency in the long run or compare total model size vs. baselines like L2P or ST-Prompt.

Dependence on CLIP: The method relies heavily on frozen CLIP features. While common, this raises questions about generalizability to non-pretrained or non-ViT backbones. A brief experiment with a non-CLIP backbone (e.g., VideoMAE, TimeSformer) would strengthen claims about framework generality.

Temporal Anchor Construction: During inference, TD-MoE uses class-wise mean temporal features (“anchors”) from the current task. This assumes access to current-task data during inference to build anchors. Clarification is needed: are anchors computed once after training and stored? If so, does this constitute a form of “exemplar” (albeit aggregated)? The paper claims “exemplar-free,” but storing per-class statistics could be seen as a lightweight memory buffer.

Kinetics-400 Results: On Kinetics-400 (Table 3), StPR achieves higher final accuracy but also higher BWF than some exemplar-based methods (e.g., CSTA). This trade-off should be discussed—does higher accuracy come at the cost of more forgetting on earlier tasks?

**Questions:**

Scalability: How does StPR scale to 50+ incremental tasks? Is there a plan to compress or merge experts (e.g., via expert pruning or clustering) to avoid linear parameter growth?

Anchor Storage: Are the temporal anchors  stored permanently after training each task? If yes, how much memory do they consume? Does this violate the “exemplar-free” claim, even if minimally?

Backbone Generality: Have you tested StPR with non-CLIP backbones (e.g., VideoSwin, TimeSformer)? Is the performance gain primarily due to CLIP’s strong priors or the StPR mechanism itself?

Task Boundaries: TD-MoE assumes known task boundaries during training (to assign experts). How would StPR handle task-agnostic or online VCIL where task boundaries are unknown?

Comparison to Prompt-Based Methods: ST-Prompt† (Pei et al., 2023) also uses spatiotemporal prompting. How does TD-MoE fundamentally differ? Is the gain due to routing or the disentanglement design?

---

> ### Author Response · Authors · 2025-11-22
> **Response to Reviewer VZo2: Part 1 (Q1, W1)**
>
> Thank you very much for your positive recognition of our work. Your suggestions and questions are highly valuable and greatly help us further improve the paper. We address each point in detail below. The additional experiments and discussions will be updated in the camera-ready version.
>
> **Q1 \& W1:**
> Regarding the parameter efficiency and the exploration of mitigating this issue, we conducted the following experiments and provide the corresponding explanation:
>
> Thank you for the suggestion. Indeed, the growth of experts with the number of tasks is a common limitation of MoE-based task-specific design continual learning [1]. Since our motivation is to leverage temporal information for routing to mitigate forgetting, **our main contribution lies in disentangling temporal information for expert routing**. However, we still explore some preliminary studies on the design of experts. As shown in **Table 1** (percentages denote module–to–CLIP parameter ratios), reducing expert depth lowers parameters and slightly affects temporal modeling, but even a single-layer expert with TD-MoE still outperforms state-of-the-art performance ($\overline{\mathrm{Acc}} = 84.75$).
>
> To minimize the number of experts as much as possible, we further evaluate an efficient design: we tested a simple selective-expert strategy by keeping only latest five experts **Table 2**. Because the inputs of all experts and temporal anchors rely on shared spatial features, newly trained experts naturally retain part of the old knowledge. Under this selection mechanism, the standard incremental setting (5×10) still exceeds the SOTA performance ($\overline{\mathrm{Acc}} = 84.75$). This demonstrates that selective expert retention is feasible.
> Finally, we compare our selective-expert scheme with the open-source L2P in **Table 3** (percentages denote module–to–CLIP parameter ratios). L2P’s prompts grow linearly with the number of classes, whereas our selecting method avoids linear growth and still achieves notably higher performance, with the best results obtained when all experts are retained.
>
> In future work, we plan to further explore more principled expert-selection mechanisms and investigate how to transfer knowledge across experts to reduce the performance gap when using a fixed number of experts.
>
> **Table 1: Performance of different layers on UCF101 (10×5s).**
>
> |               | $\overline{\mathrm{Acc}} \uparrow$ | Acc $\uparrow$ | Parameters |
> |---------------|------------------------------------|----------------|------------|
> | 1 layer       |         91.29                      |         83.75  |  3M(2.01\%)|
> | 2 layers      |         92.72                      |         85.29  |  6M(4.02\%)|
> | 3 layers      |         94.67                      |         88.85  |  9M(6.03\%)|
>
> **Table 2: Performance of selecting latest 5 experts.**
>
> |                |  Select latest 5 | $\overline{\mathrm{Acc}} \uparrow$ |
> |----------------|------------------|------------------------------------|
> | UCF101 (5×10s) |      √           |                88.77               |
> | UCF101 (5×10s) |      ×           |                92.13               |
> | UCF101 (2×25s) |      √           |                82.15               |
> | UCF101 (2×25s) |      ×           |                88.52
>
> **Table 3: Comparison with L2P after selecting the latest 5 experts on UCF101.**
>
> |                |           Method         | $\overline{\mathrm{Acc}} \uparrow$ |      Parameter Efficiency     |
> |----------------|--------------------------|------------------------------------|-------------------------------|
> | UCF101 (5×10s) |      L2P                 |                80.09               |      0.12M (0.08\%) Per Class |
> | UCF101 (5×10s) |      Select Latest 5     |                88.77               |      9M (6.03\%) Per Task     |
> | UCF101 (5×10s) |      without Selecting   |                92.13               |      9M (6.03\%) Per Task     |
> | UCF101 (2×25s) |      L2P                 |                78.58               |      0.12M (0.08\%) Per Class |
> | UCF101 (2×25s) |      Select Latest 5     |                82.15               |      9M (6.03\%) Per Task     |
> | UCF101 (2×25s) |      without Selecting   |                88.52               |      9M (6.03\%) Per Task     |
>
> [1]. Wang Y, Zhou D W, Ye H J. Integrating Task-Specific and Universal Adapters for Pre-Trained Model-based Class-Incremental Learning[C]//Proceedings of the IEEE/CVF International Conference on Computer Vision. 2025: 806-816.

---

> > ### Author Response · Authors · 2025-11-22
> > **Response to Reviewer VZo2: Part 2 (Q2, Q3, Q4, W2, W3)**
> >
> > **Q2 \& W3:**
> > Regarding whether the use of anchors violates the exemplar-free setting, we provide the following explanation:
> >
> > Current exemplar-based VCIL methods typically store raw video samples, which raises serious concerns regarding memory cost and privacy. In contrast, our anchor-based design stores only a single vector per class ($\mathbb{R}^{1\times 512}$, **1KB** per class). Compared to RGB frames (as the SM [2] method stores **0.61 GB** of video-frame exemplars), the memory footprint of such a vector is negligible, and it contains no identifiable visual content, thus avoiding privacy issues entirely. Therefore, we do not regard these anchors as exemplars; rather, they are lightweight class statistics that introduce neither memory burden nor privacy concerns.
> >
> > [2]. Zhao H, Qin X, Su S, et al. When video classification meets incremental classes[C]//Proceedings of the 29th ACM International Conference on Multimedia. 2021: 880-889.
> >
> > **Q3 \& W2:**
> > To further demonstrate the generalization ability of our method, we conducted ablation studies using ResNet-34 as the backbone, as shown in **Table 4**. Even without the strong priors provided by Transformer architectures, our modules still yield clear performance improvements. This indicates that our method does not rely on backbone-specific priors and exhibits strong transferability across architectures.
> >
> > **Table 4: Ablation of FSSD and TD-MoE on UCF101 using a ResNet-34 backbone.**
> >
> > |                        | $\overline{\mathrm{Acc}} \uparrow$  (10 × 5s ) |$ Acc \uparrow$  (10 × 5s )| $\overline{\mathrm{Acc}} \uparrow$  (5 × 10s ) | $Acc \uparrow$  (5 × 10s )|
> > |------------------------|------------------------------------------------|---------------------------|------------------------------------------------|---------------------------|
> > | Sequential Fine-Tuning |         71.05                                  |          65.66            |         69.17                                  |         62.97             |
> > | + FSSD                 |         72.92                                  |          66.95            |         70.55                                  |         64.42             |
> > | + FSSD and TD-MoE      |         76.82                                  |          71.95            |         74.83                                  |         70.82             |
> >
> >
> > **Q4:**
> > The following discusses TD-MoE in scenarios with unknown task boundaries or in online incremental settings:
> >
> > TD-MoE routes samples based on their similarity to the class-wise temporal anchors. This mechanism does not inherently require explicit task boundaries. Even in task-agnostic or online VCIL settings, we can continually accumulate temporal features for each class and update the anchors accordingly. Moreover, task grouping can be performed heuristically by controlling how many samples (or classes) each expert should handle, based on model capacity. Since our routing is soft and similarity-based, a class can activate multiple experts if needed, and the router will naturally assign higher weights to experts whose anchors align better with the sample. This allows TD-MoE to adapt to settings where task boundaries are unknown or not strictly defined.

---

> ### Author Response · Authors · 2025-11-22
> **Response to Reviewer VZo2: Part 3 (Q5, W4)**
>
> **Q5:**
> The following explains the distinction between our method and ST-Prompt†:
>
> (1) **Different Experts.** ST-Prompt† follows the L2P prompt-based fine-tuning approach from static images and inserts prompts along three dimensions (spatial, temporal, and aggregation) into video frames. In contrast, we apply a multi-head self-attention mechanism to all extracted video frame features to capture global inter-frame information. (2) **Different Routing.** ST-Prompt† uses K-Means clustering based on the extracted and stored prompt features to route samples to different experts, which is a routing mechanism more suited for static images. Our TD-MoE, however, disentangles temporal features and routes samples to experts based on different temporal characteristics between classes, making it more suitable for video class-incremental learning.
>
> **Table 5** results show that, using the same CLIP backbone, our method outperforms ST-Prompt† by a significant margin on the UCF101, HMDB51, and SSv2 datasets.
>
> **Table 5: Performance of ST-Prompt† and Our StPR Method on UCF101, HMDB51 and SSv2.**
> |             |  UCF101 |  UCF101 |  UCF101 | HMDB51    | HMDB51    |  SSv2   |    SSv2   |
> |-------------|---------|---------|---------| ----------|-----------|---------|-----------|
> |             | 10 × 5s | 5 × 10s | 2 × 25s |  5 × 5s   |  1 × 25s  | 10 × 9s |  5 × 18s  |
> | ST-Prompt†  | 84.75   | 85.54   | 85.67   |  60.14    |  60.54    |  39.98  |   35.44   |
> | StPR (ours) |**94.67**|**92.13**|**88.52**| **68.12** | **67.01** |**40.79**| **37.30** |
>
> **W4:**
> The following explains the forgetting issue on the Kinetics-400 dataset:
>
> In continual learning, both average accuracy and final accuracy consider the model's forgetting of old tasks and learning of new tasks. Higher average and final accuracies do not come at the cost of increased forgetting, but rather indicate a better balance between stability and plasticity. Storing samples is essentially a strategy that mitigates forgetting, similar to joint training, which is why exemplar-based methods naturally exhibit lower forgetting. However, even compared to exemplar-based methods such as CSTA, which store samples from previous tasks, our method demonstrates significant improvements in both average and final accuracy, proving the superiority of our model.
>
> We once again thank you for your valuable comments and suggestions. We hope our responses have addressed your concerns and would be happy to clarify any remaining questions; we would greatly appreciate it if you could consider raising the score in light of these responses.

---

### Official Review · Reviewer_vccK · 2025-10-30

**Soundness:** 2
**Presentation:** 3
**Contribution:** 2
**Rating:** 4
**Confidence:** 4

**Summary:**

The paper addresses the challenges of catastrophic forgetting and spatiotemporal modeling in Video Class-Incremental Learning (VCIL) by proposing an exemplar-free framework named StPR. Its contributions are twofold: 1）It proposes a Frame-Shared Semantics Distillation method (FSSD) that preserves frame-shared, semantically aligned spatial channels through semantic importance-aware regularization, optimizing the stability-plasticity trade-off in continual learning; 2）It designs a Temporal Decomposition based Mixture-of-Experts strategy (TD-MoE) that decomposes spatiotemporal features and uses temporal dynamics for expert combination, enabling task-id-free and dynamic adaptation.

**Strengths:**

1) The paper proposes an exemplar-free setting, which avoids the memory and privacy issues associated with storing historical data, making it more practical for real-world applications.
2) The theoretical analysis is substantial, providing derivations for the channel importance metrics (Fisher Information, classification contribution) used in FSSD.
3) The experimental analysis is thorough, validating the method's effectiveness across multiple datasets and various task partitions, and includes comprehensive ablation studies and analysis.

**Weaknesses:**

1) Line 70: "reuses these decomposed components to enhance the model’s ability to adapt continually, thereby reducing forgetting without storing extensive exemplars." Is this statement problematic, as the subsequent text does not mention reusing these decomposed components? Furthermore, the claim that decomposed components help enhance adaptability is also debatable.

2) Why are channels with high semantic importance preserved? The semantic importance from a previous session does not necessarily guarantee importance in subsequent sessions.

3) If the mixture-of-experts system can effectively address catastrophic forgetting and adaptive learning, why is the distillation module necessary? Moreover, mixture-of-experts has already been used in class-incremental learning[1], and no significant distinction from your method is evident.
[1] Yu J, Zhuge Y, Zhang L, et al. Boosting continual learning of vision-language models via mixture-of-experts adapters[C]//Proceedings of the IEEE/CVF Conference on Computer Vision and Pattern Recognition. 2024: 23219-23230.

4) On the Kinetics-400 dataset, there is a lack of comparison with results from the latest top-tier conference papers.

5) It is recommended to supplement experiments with longer video sequences (e.g., 16 or 32 frames) or higher frame rate inputs to analyze TD-MoE's capability in long-term temporal modeling and its impact on computational cost.

**Questions:**

Seen above.

---

> ### Author Response · Authors · 2025-11-22
> **Response to Reviewer vccK： Part 1 (Q1, Q2)**
>
> Thank you very much for your valuable comments and suggestions, which are highly helpful for improving our work. We address your concerns point-by-point below. The additional experiments and discussions will be updated in the camera-ready version.
>
> **Q1:**
> We provide the following explanation regarding the reuse of decoupled features and the model's adaptability:
>
> We reuse two types of disentangled features: 1) the **semantic importance** used in FSSD, and 2) the **temporal information** used in TD-MoE routing. For FSSD, the estimated semantic importance serves as adaptive distillation weights, preserving channels crucial for old tasks while relaxing constraints on less important ones to better learn new tasks. This achieves a stronger stability–plasticity balance. For TD-MoE, class-wise temporal features are used as temporal anchors. During inference, each sample is matched to these anchors, and experts whose temporal patterns align more closely receive higher routing weights. This provides adaptive, temporally guided routing that effectively mitigates forgetting. Together, these adaptive distillation and routing mechanisms enhance the model’s overall adaptability in VCIL. As shown in **Table 1**, the results show that adding either component individually leads to significant performance gains, which supports our claim.
>
> **Table 1: Acc Ablation of FSSD and TD-MoE.**
>
> | Method           | UCF101($5 \times 10$s) | HMDB51($5 \times 5$s) | UCF101($10 \times 5$s) | HMDB51($25 \times 1$s) |
> |------------------|------------------------|-----------------------|------------------------|------------------------|
> |Adapter Tuning    |         74.23          |         48.99         |           74.71        |           52.92        |
> | +FSSD            |         77.55          |         53.23         |           77.63        |           54.63        |
> | +TD-MoE          |         83.07          |         57.47         |           88.03        |           64.71        |
> | +FSSD and TD-MoE |         **85.79**      |         **63.04**     |           **88.85**    |           **69.61**    |
>
> **Q2:**
> We provide the following explanation regarding why semantic important channels should be preserved and the importance of the gap between new and old tasks' channels:
>
> Our proposed FSSD aims to **adaptively mitigate forgetting in the spatial feature extractor**. The semantic importance we introduce consists of two components: semantic sensitivity and classification score. The former captures the channel’s stability for a given task, while the latter reflects its discriminative contribution to the current task. Together, these two metrics allow us to reliably assess the importance of each channel. Moreover, for a new task, channels that are important for old tasks may be unimportant for the new one. In such cases, imposing a stronger constraint on these channels preserves old knowledge without overly hindering new-task learning. Conversely, if a channel is important for both old and new tasks, we regard this as strong cross-task correlation. A tighter constraint on such channels, together with the classification loss, encourages the model to leverage old-task knowledge when learning the new task, while shifting the learning of new information to channels less critical for previous tasks. This achieves an adaptive balance between preservation and plasticity.

---

> > ### Author Response · Authors · 2025-11-22
> > **Response to Reviewer vccK： Part 2 (Q3)**
> >
> > **Q3:**
> > We provide the following explanation regarding the distinct problems addressed by the FSSD and TD-MoE modules, and how they differ from the work in [1]:
> >
> > FSSD and TD-MoE address different forms of forgetting in our framework and are complementary to each other. FSSD uses channel-wise semantic sensitivity to adaptively preserve spatial feature knowledge, while TD-MoE leverages disentangled temporal information as expert routing signals to retain temporal knowledge across tasks. Specifically, FSSD mitigates the forgetting of spatial features. Since the spatial encoder serves as a shallow network for extracting frame-level spatial information and its forward error is relatively small, we improve parameter efficiency by sharing this module across all tasks. However, this also means that its feature space inevitably drifts toward the new task during training. FSSD suppresses this drift while still allowing effective learning of the new task. TD-MoE, in contrast, targets the forgetting of temporal information. The temporal anchors used in TD-MoE are computed based on the spatial representations produced by the shared encoder, meaning that temporal stability still depends on the stability of the underlying spatial features. Thus, FSSD is complementary to TD-MoE, and both are necessary for maintaining performance across sessions. As shown in **Table 1**, this results further confirm that adding FSSD on top of TD-MoE (StPR) effectively resolves this issue, as the performance consistently improves compared to using TD-MoE alone (w/o FSSD).
> >
> > Our method is foundamentally different from existing method [1]. Our approach differs in the following aspects: (1) **Different Experts.** As the method in [1] addresses the class-incremental learning for static images, their experts are embedded adapter modules at the static image level to extract image features. However, we focuses on the more complex challenge of video class-incremental learning. Our experts are added after the spatial feature extraction with multi-head self-attention layers, aimed at capturing spatiotemporal features across frames. (2) **Different Routing Design.** The method in [1] relies on the CLS token computed from image features for routing, whereas, as discussed in our introduction, video class-incremental learning requires modeling intrinsic temporal dynamics. Our method uses disentangled temporal information as the routing mechanism for experts, which is more suitable for video incremental learning. Additionally, we demonstrate through experiments that our method outperforms [1] in the video class-incremental task. As shown in **Table 2**, our TD-MoE method clearly outperforms Adapter-MoE, which is designed following [1], across different datasets and settings. This demonstrates the effectiveness of our method and highlights the key differences.
> >
> > **Table 2: Acc Performance of Adapter-MoE and TD-MoE.**
> >
> > | Method         | UCF101($5 \times 10$s) | HMDB51($5 \times 5$s) | UCF101($10 \times 5$s) | HMDB51($25 \times 1$s) |
> > |----------------|------------------------|-----------------------|------------------------|------------------------|
> > | Adapter-MoE    |         83.26          |         61.99         |           84.31        |           65.57        |
> > | TD-MoE         |         **85.79**      |         **63.04**     |           **88.85**    |           **69.61**    |
> >
> > [1] Yu J, Zhuge Y, Zhang L, et al. Boosting continual learning of vision-language models via mixture-of-experts adapters[C]//Proceedings of the IEEE/CVF Conference on Computer Vision and Pattern Recognition. 2024: 23219-23230.

---

> ### Author Response · Authors · 2025-11-22
> **Response to Reviewer vccK： Part 3 (Q4, Q5)**
>
> **Q4:**
> Regarding the lack of recent top-tier conference papers on the Kinetics-400 dataset, after further investigation, we found the PIVOT method [2] presented at CVPR 2023. PIVOT uses prompt-based fine-tuning of the spatiotemporal encoder and trains using stored samples in a three-stage process. However, it is somewhat puzzling that the reported results without prompts outperformed those with prompts. We speculate that this may be due to prompts, as feature embeddings, not effectively capturing temporal relations between video frames, potentially leading to task boundary confusion. We compared the results of PIVOT with and without prompts to ours (**Table 3**). On 10 tasks, our method outperforms PIVOT with prompts in terms of accuracy. On 20 tasks, due to its reliance on a large number of stored old samples, this training approach is closer to joint training, which results in better performance over a longer period. Both the 10s and 20s settings show significantly lower forgetting than PIVOT. Additionally, while PIVOT relies heavily on video sample replay to reduce forgetting, which introduces significant memory pressure and privacy concerns, our exemplar-free method avoids both issues while achieving better performance on 10 tasks than PIVOT.
>
> Regarding VCIL works from 2024 and 2025, our research shows that current methods still focus primarily on smaller datasets such as UCF101, HMDB51, and SSv2. On these datasets, our method significantly outperforms the SOTA.
>
> **Table 3: Performance of StPR and PIVOT on Kinetics-400.**
>
> | Method               | 10s: Acc               | 10s: BWF              | 20s: Acc               |  20s: BWF              |        Memory           |
> |----------------------|------------------------|-----------------------|------------------------|------------------------|-------------------------|
> | PIVOT [2]            |         55.13          |         26.50         |           55.04        |           26.41        |        4000MB           |
> | PIVOT w/o prompts [2]|         58.61          |         20.78         |           57.51        |           21.98        |        4000MB           |
> | StPR                 |         57.83          |         14.01         |           53.95        |           15.09        |        0                |
>
> [2] Villa A, Alcázar J L, Alfarra M, et al. Pivot: Prompting for video continual learning[C]//Proceedings of the IEEE/CVF Conference on Computer Vision and Pattern Recognition. 2023: 24214-24223.
>
> **Q5:**
> To investigate the ability of TD-MoE to capture temporal information with varying frame numbers in the sampled data, we conducted the following experiments and provide the corresponding explanation:
>
> To analyze TD-MoE’s ability to handle different degrees of temporal information, we conducted additional experiments on UCF101 with input clips containing sparse (2, 4 frames), adequate (8, 16 frames), and redundant (32 frames) temporal cues. The results are shown in **Table 4** (percentages denote module–to–CLIP FLOPs ratios). We observe that performance increases as more temporal information becomes available, peaking at 8–16 frames. Notably, even when the frame count is doubled to 32, the accuracy remains stable, indicating that TD-MoE can effectively exploit longer temporal sequences without being negatively affected by redundancy. In terms of computational cost, the FLOPs of the TD-MoE experts remain extremely small (**0.01\%–0.12\%**) relative to the backbone, demonstrating that TD-MoE introduces negligible overhead even when modeling longer sequences. Together, these results confirm that TD-MoE maintains strong long-term temporal modeling capability while keeping the computational cost minimal.
>
> **Table 4: Performance of TD-MoE with Different Frame Counts on UCF101 (10×5s).**
>
> | Frames         | $\overline{\mathrm{Acc}} \uparrow$ | Acc $\uparrow$ | BWF $\downarrow$ | FLOPs (GFLOPs) $\downarrow$ |
> |----------------|------------------------------------|----------------|------------------|-----------------------------|
> | 2              |               91.29                |     83.75      |       8.91       |             0.0285 (0.01\%) |
> | 4              |               92.75                |     85.29      |       8.60       |             0.0474 (0.02\%) |
> | 8              |               94.67                |     88.85      |       6.31       |             0.0854 (0.03\%) |
> | 16             |               94.50                |     89.54      |       5.58       |             0.1613 (0.06\%) |
> | 32             |               93.39                |     89.48      |       4.72       |             0.3131 (0.12\%) |
>
> We once again thank you for your valuable comments and suggestions. We hope our responses have addressed your concerns and would be happy to clarify any remaining questions; we would greatly appreciate it if you could consider raising the score in light of these responses.

---

> > ### Comment · Reviewer_vccK · 2025-11-28
> >
> > I would like to thank the authors for their feedback. Considering that the author's response has addressed some of my confusion, I have decided to rasie the score to 6 points.

---

> > > ### Author Response · Authors · 2025-11-30
> > >
> > > We sincerely thank you for your careful evaluation of our work and thoughtful, constructive feedback, as well as your recognition of our contributions. We will continue to refine and improve our work based on your valuable suggestions.

---

### Comment · Area_Chair_Hs4F · 2025-11-27

Dear Reviewers,

As we enter the discussion phase, I strongly encourage you to read the authors' rebuttal carefully and acknowledge their effort. Silence is the worst outcome for an author. Even if the rebuttal does not change your final rating, a brief response explaining why the concerns remain unaddressed is crucial for a fair process. Please help us make an informed decision by engaging in a constructive dialogue.

AC

---

### Author Response · Authors · 2025-11-30
**Global Response (Part 1 of 2)**

Our Global Response includes the following points:
1. **Changes in scores during the discussion period**;

2. **Introduction to our paper**;

3. **Our contributions**;

4. **Summary of strengths mentioned by the reviewers**;

5. **Summary of the main concerns raised by the reviewers**;

6. **Our rebuttal**.

___

**1. Changes in scores during the discussion period:**


During the discussion phase, **Reviewer vccK indicated raising the score to 6.**
___

**2. Introduction to our paper:**

Our work proposes the StPR algorithm for Video Class-Incremental Learning (**VCIL**) in an **exemplar-free setting**. Unlike static image class-incremental learning, the key challenge in VCIL is "*mitigating catastrophic forgetting while effectively leveraging frame-shared semantics and temporal dynamics to incrementally learn new categories.*" To address this, we designed two modules:

(1) **Frame-Shared Semantic Distillation (FSSD):** Through detailed theoretical derivation, we estimate the importance of semantic channels in the video space for different tasks, encouraging the model to strengthen the constraints on important channels for old tasks, while relaxing the constraints on less important channels to better learn new tasks, thus achieving a better balance between stability and plasticity.

(2) **Temporal-Decomposition-based Mixture-of-Experts (TD-MoE):** Through experimental verification and theoretical approximation, we decouple the temporal information in videos, preserving temporal anchor points for each category so that the model can utilize temporal information during inference to reduce confusion in decision boundaries between tasks.

___

**3. Our contributions:**

(1) For videos, we propose a **novel method to capture inherent spatial and temporal features** and use this spatiotemporal information to achieve a better balance between stability and plasticity in incremental settings. (2) Our **exemplar-free setting** greatly alleviates memory pressure and eliminates privacy concerns, making it more practical. (3) Our algorithm **achieves state-of-the-art (SOTA) performance** on four benchmark public datasets, outperforming exemplar-based methods.

___

**4. Summary of strengths mentioned by the reviewers:**

 (1). The **exemplar-free setting** is practically significant.

 (2). The proposed method is **novel and interesting**.

 (3). **Theoretical analysis is solid.**

 (4). **Comprehensive, extensive, and effective experiments**.

 (5). **Clear writing** and well-organized structure.

___

**5. Summary of the main concerns raised by the reviewers:**

Q1. **Technical details.**

Q2. Generalization across **different backbones** and plug-in nature of the FSSD module in **different baselines**.

Q3. **Parameter efficiency.**

Q4. **Comparison with other methods.**

Q5. Discussion on **different settings**.

---

> ### Author Response · Authors · 2025-11-30
> **Global Response (Part 2 of 2)**
>
> **6. Our rebuttal:**
>
> **Q1: Technical details:**
> We have **fully explained and analyzed:**
>
> (1) Reuse of features,
>
> (2) The meaning of FSSD channel importance,
>
> (3) The explanation of temporal anchors not being exemplars,
>
> (4) The motivation and complementarity of FSSD and TD-MoE, with supporting ablations.
>
> **Q2:**
> (1) **Generalization across different backbones:** In response to the concern about our heavy reliance on pre-trained CLIP priors and generalization across other backbones, we directly use a non-transformer architecture, Resnet34, which has no strong priors, as our backbone and validate the effectiveness of our modules on it.
>
> (2) **Plug-in nature of the FSSD module:** Based on open-source implementations, we directly add our FSSD module to the TCD [1] baseline and the experimental results validate its plug-in nature and effectiveness.
>
> **Q3:**
> **Parameter efficiency of TD-MoE:** Although our **main contribution** lies in **disentangling temporal information for expert routing**, we still explore some preliminary studies on the design of experts.
>
> (1) We experimented with **reducing the parameters of experts** and found that the performance still **achieves SOTA.**
>
> (2) We **fixed the expert pool size to 5 experts** and found that, under the classic 10s setting, the performance of our algorithm **is still SOTA**. We report the training parameters and computational costs and compare the parameter efficiency with the classical L2P [2] method. We plan to further explore knowledge transfer between experts to build a more powerful expert pool.
>
> **Q4: Comparison with other methods:**
>
> (1) We conducted detailed analysis and experimental comparisons with Adapter-MoE [3] and ST-Prompt† [4] and demonstrated that our method **differs and outperforms them**.
>
> (2) We compared the performance of our algorithm with PIVOT [5], CoSTEP [6], and CRAM [7], and **further validated our method on the SM [8] setting**, showing that our method **is superior to these methods**.
>
> (3) We **reproduced the EWC [9]** algorithm and performed experimental and **analyses with our FSSD algorithm**, showing that **our FSSD method is better suited for VCIL tasks**.
>
> **Q5: Discussion on different settings:**
>
> (1) We validated **TD-MoE's capability in long-term temporal modeling** and its impact on **computational cost** across video samples with 2, 4, 8, 16, and 32 frames, and reported the corresponding FLOPs.
>
> (2) We added ablation experiments on a SSv2 for the TD-MoE module, proving its ability to **capture temporal information on temporally challenging datasets.**
>
> (3) We thoroughly discussed the feasibility of our solution during training with **unknown task boundaries and even in online continual learning tasks.**
>
> (4) Regarding the **choice of different fine-tuning settings**, we addressed the reviewer's concern by citing prior work, specifically SSIAT [10].
>
> We sincerely appreciate your dedicated and outstanding review work!
>
> [1] Park J, Kang M, Han B. Class-incremental learning for action recognition in videos[C]//Proceedings of the IEEE/CVF international conference on computer vision. 2021: 13698-13707.
>
> [2] Wang Z, Zhang Z, Lee C Y, et al. Learning to prompt for continual learning[C]//Proceedings of the IEEE/CVF conference on computer vision and pattern recognition. 2022: 139-149.
>
> [3] Yu J, Zhuge Y, Zhang L, et al. Boosting continual learning of vision-language models via mixture-of-experts adapters[C]//Proceedings of the IEEE/CVF Conference on Computer Vision and Pattern Recognition. 2024: 23219-23230.
>
> [4] Pei Y, Qing Z, Zhang S, et al. Space-time prompting for video class-incremental learning[C]//Proceedings of the IEEE/CVF International Conference on Computer Vision. 2023: 11932-11942.
>
> [5] Villa A, Alcázar J L, Alfarra M, et al. Pivot: Prompting for video continual learning[C]//Proceedings of the IEEE/CVF Conference on Computer Vision and Pattern Recognition. 2023: 24214-24223.
>
> [6] Zou X, Ma W, Zhao S. Learning Conditional Space-Time Prompt Distributions for Video Class-Incremental Learning[C]//Proceedings of the Computer Vision and Pattern Recognition Conference. 2025: 4862-4873.
>
> [7] Mall S, Henriques J F. CRAM: Large Scale Video Continual Learning with Bootstrapped Compression[C]//Proceedings of the IEEE/CVF International Conference on Computer Vision. 2025: 15045-15055.
>
> [8] Zhao H, Qin X, Su S, et al. When video classification meets incremental classes[C]//Proceedings of the 29th ACM International Conference on Multimedia. 2021: 880-889.
>
> [9] Kirkpatrick J, Pascanu R, Rabinowitz N, et al. Overcoming catastrophic forgetting in neural networks[J]. Proceedings of the national academy of sciences, 2017, 114(13): 3521-3526.
>
> [10] Tan Y, Zhou Q, Xiang X, et al. Semantically-shifted incremental adapter-tuning is a continual vitransformer[C]//Proceedings of the IEEE/CVF Conference on Computer Vision and Pattern Recognition. 2024: 23252-23262.

---

### Meta-Review · Area_Chair_GG4g · 2025-12-27

**Summary:**

The paper proposes StPR, an exemplar-free framework for video class-incremental learning that mitigates catastrophic forgetting by preserving frame-shared semantics and dynamically routing temporal experts, achieving strong performance across multiple video benchmarks.

Across reviewers, the main concerns initially centered on limited conceptual novelty relative to prior VCIL methods, the unclear necessity and distinction of FSSD and TD-MoE from existing regularization, prompting, and MoE-based approaches, scalability issues due to linear parameter growth, heavy reliance on CLIP features, ambiguity surrounding the “exemplar-free” claim, insufficient analysis of forgetting and temporal modeling in challenging settings, and missing or outdated comparisons with recent state-of-the-art methods.

Despite these concerns, reviewers consistently acknowledged the importance of the problem, the strong motivation of addressing VCIL, and the overall soundness of the proposed approach, with comprehensive experiments showing notable improvements over baselines. During rebuttal, the authors effectively addressed the raised issues through additional well-designed experiments and clarifications, which the Area Chair finds sufficient to resolve the reviewers’ major concerns.

Overall, the contributions outweigh the remaining limitations, and the additional experimental evidence substantially strengthens the paper. The Area Chair therefore recommends acceptance as a poster, believing that the exemplar-free design is a valuable and timely contribution to video class-incremental learning and will be useful for a broad range of continual learning applications.

**Reviewer Concerns:**

Reviewer vccK questioned the conceptual soundness and novelty of FSSD and TD-MoE, their distinction from prior MoE-based CIL methods, the lack of comparisons with recent state-of-the-art approaches (particularly on Kinetics-400), and the absence of evaluations on longer or more temporally challenging video settings.

Reviewer VZo2 raised concerns about the scalability and linear parameter growth of TD-MoE, heavy reliance on CLIP features and limited backbone generality, ambiguity surrounding temporal anchor storage and the “exemplar-free” claim, assumptions about known task boundaries, and missing comparisons with prompt-based VCIL methods.

Reviewer SftT focused on the linear scaling of memory and computation with the number of tasks, insufficient analysis of forgetting on temporally challenging datasets, and unclear distinctions between FSSD and established importance-based regularization methods such as EWC.

Reviewer s53Z questioned the standalone effectiveness and generality of FSSD as a plug-in module, noted missing comparisons with prior and recent VCIL methods, highlighted presentation issues, and requested clearer positioning among different parameter-efficient fine-tuning strategies in VCIL.

The authors addressed these concerns comprehensively through substantial additional experiments and analyses. They provided expanded comparisons with newly added baselines, detailed ablations on video length (2–32 frames), and quantitative analyses of memory storage, computational cost, and parameter efficiency. Technical clarifications were added regarding feature reuse, the interpretation of FSSD channel importance, the non-exemplar nature of temporal anchors, and the complementary roles of FSSD and TD-MoE, supported by targeted ablations. To address generalization concerns, the authors validated the framework on a non-CLIP, non-transformer backbone (ResNet-34) and demonstrated the plug-in effectiveness of FSSD on an external VCIL baseline. Scalability issues were examined via reduced-size experts, fixed expert pools, and comparisons against L2P, showing competitive parameter efficiency. The authors also conducted new comparisons with prompt-based and MoE-based VCIL methods (e.g., ST-Prompt†, Adapter-MoE), recent state-of-the-art approaches (e.g., PIVOT, CoSTEP, CRAM), and EWC, and added evaluations on temporally challenging datasets such as SSv2. Finally, they discussed extensions to unknown task boundaries and online continual learning settings, and clarified design choices by grounding them in prior work.

Overall, the additional experiments and analyses sufficiently address the reviewers’ concerns and substantially strengthen the paper’s empirical and conceptual foundations.

**Reviewer Scores:**

The reviewers are likely to increase their initial scores in light of the comprehensive and well-designed experiments provided by the authors during rebuttal; notably, Reviewer vccK has already indicated raising their score to 6 even prior to the ICLR incident.

---

### Decision · Program_Chairs · 2026-01-26

Accept (Poster)